# FairICP: Encouraging Equalized Odds via Inverse Conditional Permutation

**Yuheng Lai** [1]  **Leying Guan** [2]

## Abstract

*Equalized odds*, an important notion of algorithmic fairness, aims to ensure that sensitive variables, such as race and gender, do not unfairly influence the algorithm's prediction when conditioning on the true outcome. Despite rapid advancements, current research primarily focuses on equalized odds violations caused by a single sensitive attribute, leaving the challenge of simultaneously accounting for multiple attributes under-addressed. We bridge this gap by introducing an in-processing fairness-aware learning approach, FairICP, which integrates adversarial learning with a novel inverse conditional permutation scheme. FairICP offers a flexible and efficient scheme to promote equalized odds under fairness conditions described by complex and multi-dimensional sensitive attributes. The efficacy and adaptability of our method are demonstrated through both simulation studies and empirical analyses of real-world datasets.

## 1. Introduction

Machine learning models are increasingly important in aiding decision-making across various applications. Ensuring fairness in these models—preventing discrimination against minorities or other protected groups—remains a significant challenge (Mehrabi et al., 2021). To address different needs, several fairness metrics have been developed in the literature (Mehrabi et al., 2021; Castelnovo et al., 2022). The *equalized odds* metric defines fairness by requiring that the predicted outcome $\hat{Y}$ provides the same level of information about the true response $Y$ across different sensitive attribute(s) $A$ (e.g. gender/race/age) (Hardt et al., 2016):

$$\hat{Y} \perp\!\!\!\perp A \mid Y. \tag{1}$$

Encouraging *equalized odds* is more challenging due to $Y$-conditioning compared to encouraging *demographic parity*, another common fairness notion that emphasizes *equity* and requires $\hat{Y}$ to be independent of $A$ without conditioning on $Y$. Most existing algorithms targeting equalized odds can only handle a single protected attribute, and extending them to multi-dimensional sensitive attributes is challenging due to the well-known difficulties of estimating multi-dimensional densities (Scott, 2015). However, real-world scenarios can involve biases that arise from multiple sensitive attributes simultaneously. For example, in healthcare settings, patient outcomes can be influenced by a combination of race, gender, and age (Ghassemi et al., 2021; Yang et al., 2022). Moreover, ignoring the correlation between multiple sensitive attributes can lead to *fairness gerrymandering* (Kearns et al., 2018), where a model appears fair when considering each attribute separately but exhibits unfairness when attributes are considered jointly.

To address these limitations, we introduce *FairICP*, a flexible fairness-aware learning scheme that encourages equalized odds for complex sensitive attributes. Our method leverages a novel *Inverse Conditional Permutation (ICP)* strategy to generate conditionally permuted copies $\tilde{A}$ of sensitive attributes $A$ given $Y$ without the need to estimate the multi-dimensional conditional density and encourages equalized odds via enforcing similarity between $(\hat{Y}, A, Y)$ and $(\hat{Y}, \tilde{A}, Y)$. An illustration of the FairICP framework is provided in Figure 1.

Our contributions can be summarized as follows:

- **Inverse Conditional Permutation (ICP)**: We introduce the ICP strategy to efficiently generate $\tilde{A}$, as conditional permutations of $A$ given $Y$, without estimating the multi-dimensional conditional density of $A|Y$. This makes our method scalable and applicable to complex sensitive attributes.

- **Empirical Validation**: Through simulations and real-world data experiments, we demonstrate FairICP's flexibility and its superior fairness-accuracy trade-off compared to existing methods targeting equalized odds. Our results also confirm that ICP is an effective sensitive attribute resampling technique for achieving equalized odds with increased dimensions.

---

[1]Department of Statistics, University of Wisconsin-Madison [2]Department of Biostatistics, Yale University. Correspondence to: Leying Guan <leying.guan@yale.edu>.

*Proceedings of the 42$^{nd}$ International Conference on Machine Learning*, Vancouver, Canada. PMLR 267, 2025. Copyright 2025 by the author(s).

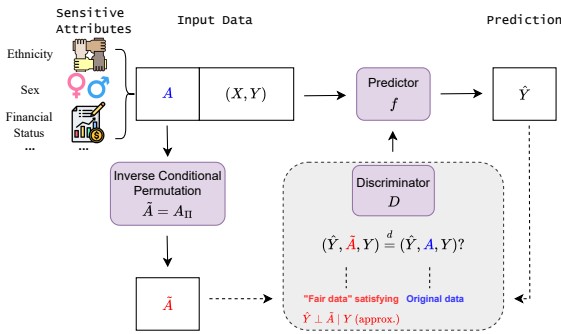

Figure 1: Illustration of the FairICP framework. $A$, $X$, and $Y$ denote the sensitive attributes, features, and labels. We generate $\tilde{A}$ as permuted copies of $A$ which satisfies $(\hat{\mathbf{Y}}, \mathbf{A}, \mathbf{Y}) \overset{d}{=} (\hat{\mathbf{Y}}, \tilde{\mathbf{A}}, \mathbf{Y})$ when the equalized odds holds, using a novel *inverse conditional permutation (ICP)* strategy, and construct a fairness-aware learning method through regularizing the distribution of $(\hat{Y}, A, Y)$ toward the distribution of $(\hat{Y}, \tilde{A}, Y)$.

**Background and Related Work** Fairness in machine learning has emerged as a critical area of research, with various notions and approaches developed to address potential biases in algorithmic decision making. These fairness concepts can be broadly categorized into three main types: (1) *group fairness* (Hardt et al., 2016), which aims to ensure equal treatment across different demographic groups; (2) *individual fairness* (Dwork et al., 2012), focusing on similar predictions for similar individuals; and (3) *causality-based fairness* (Kusner et al., 2017), which considers fairness in counterfactual scenarios. Given a fairness condition, existing fair ML methods for encouraging it can be classified into three approaches: pre-processing (Zemel et al., 2013; Feldman et al., 2015), in-processing (Agarwal et al., 2018; Zhang et al., 2018), and post-processing (Hardt et al., 2016).

Our work focuses on in-processing learning for the *equalized odds* (Hardt et al., 2016), a group fairness concept. Equalized odds requires that predictions are independent of sensitive attributes conditional on the true outcome, unlike demographic parity (Zemel et al., 2013), which demands unconditional independence. The conditional nature of equalized odds makes it particularly challenging when dealing with complex sensitive attributes that may be multidimensional and span categorical, continuous, or mixed types. Under demographic parity, there have been several methods developed for classification tasks with multiple categorical $A$ (Agarwal et al., 2018; Kearns et al., 2018; Creager et al., 2019), however, these ideas can not be trivially generalized to equalized odds. As a result, existing work on equalized odds primarily considers one-dimensional sensitive attributes, with no prior work designed to handle multidimensional continuous or mixed-type sensitive attributes. For example, Mary et al. (2019) introduces a penalty term using the *Hirschfeld-Gebelein-Rényi Maximum Correlation Coefficient* to accommodate for a continuous sensitive attribute in both regression and classification settings. Another

line of in-processing algorithms for equalized odds uses adversarial training for a single sensitive attribute (Zhang et al., 2018; Louppe et al., 2017; Romano et al., 2020; Madras et al., 2018). Our proposed FairICP is the first in-processing framework specifically designed for equalized odds with complex sensitive attributes.

**Selected Review on Metrics Evaluating Equalized Odds Violation** Reliable evaluation of equalized odds violations is crucial for comparing equalized odds learning methods and assessing model performance in real-world applications. While numerous methods have been proposed to test for parametric or non-parametric conditional independence, measuring the degree of conditional dependence for multi-dimensional variables remains challenging. We note recent progress in two directions. One direction is the resampling-based approaches (Sen et al., 2017; Berrett et al., 2020; Tansey et al., 2022). These methods allow flexible and adaptive construction of test statistics for comparisons. However, their accuracy heavily depends on generating accurate samples from the conditional distribution of $A|Y$, which can be difficult to verify in real-world applications with unknown $A|Y$. Efforts have also been made towards direct conditional dependence measures for multi-dimensional variables. Notably, Azadkia & Chatterjee (2021) proposed CODEC, a non-parametric, tuning-free conditional dependence measure. This was later generalized into the Kernel Partial Correlation (KPC) (Huang et al., 2022):

**Definition 1.1.** *Kernel Partial Correlation* (KPC) coefficient $\rho^2 \equiv \rho^2(U, V \mid W)$ is defined as:

$$\rho^2(U, V \mid W) := \frac{\mathbb{E}\left[\mathrm{MMD}^2\left(P_{U|WV}, P_{U|W}\right)\right]}{\mathbb{E}\left[\mathrm{MMD}^2\left(\delta_U, P_{U|W}\right)\right]},$$

where $(U, V, W) \sim P$ and $P$ is supported on a subset of some topological space $\mathcal{U} \times \mathcal{V} \times \mathcal{W}$, MMD is the *maximum mean discrepancy* - a distance metric between two probability distributions depending on the characteristic kernel $k(\cdot, \cdot)$ and $\delta_U$ denotes the Dirac measure at $U$.

Under mild regularity conditions (see details in Huang et al. (2022)), $\rho^2$ satisfies several good properties for any joint distribution of $(U, V, W)$ in Definition 1.1: (1) $\rho^2 \in [0, 1]$; (2) $\rho^2 = 0$ if and only if $U \perp\!\!\!\perp V \mid W$; (3) $\rho^2 = 1$ if and only if $U$ is a measurable function of $V$ given $W$. A consistent estimator $\hat{\rho^2}$ calculated by geometric graph-based methods (Section 3 in Huang et al. (2022)) is also provided in R Package KPC (Huang, 2022). KPC allows us to rigorously quantify the violation of equalized odds for multi-dimensional $A$ via $\hat{\rho^2}(\hat{Y}, A \mid Y)$, where $A$ can take arbitrary forms and response $Y$ can be continuous (regression) or categorical (classification). Additionally, $\rho^2(\hat{Y}, A \mid Y)$ has been properly normalized in $[0, 1]$ to enable direct comparisons across different models in a given problem as it can be viewed as a generalized version of squared partial correlation between

$U$ and $V$ given $W$ (see Appendix A for a simple example). In this paper, we consider KPC as our main evaluation metric of equalized odds, where its robustness is also supported by comparison with other popular metrics (e.g., DEO for classification with categorical $A$ as in Agarwal et al. (2018); Cho et al. (2020)) in Section 3.

## 2. Method

We begin by reviewing how to conduct fairness-aware learning via sensitive attribute resampling to encourage equalized odds and its challenges with complex attributes. We then introduce our proposed method, *FairICP*, which leverages the simpler estimation of $Y|A$ to perform resampling, providing theoretical insights and practical algorithms. All proofs in this section are deferred to Appendix B.

Let $(X_i, A_i, Y_i)$ for $i = 1, \ldots, n_{\mathrm{tr}}$ be i.i.d. generated triples of (features, sensitive attributes, response). Let $f_{\theta_f}(.)$ be a prediction function with model parameter $\theta_f$. While $f_{\theta_f}(.)$ can be any differentiable prediction function, we consider it as a neural network throughout this work. Let $\hat{Y} = f_{\theta_f}(X)$ be the prediction for $Y$ given $X$. For a regression problem, $\hat{Y}$ is the predicted value of the continuous response $Y$; for a classification problem, the last layer of $f_{\theta_f}(.)$ is a softmax layer and $\hat{Y}$ is the predicted probability vector for being in each class. We also denote $\mathbf{X} = (X_1, \ldots, X_{n_{\mathrm{tr}}})$, $\mathbf{A} = (A_1, \ldots, A_{n_{\mathrm{tr}}})$, $\mathbf{Y} = (Y_1, \ldots, Y_{n_{\mathrm{tr}}})$ and $\hat{\mathbf{Y}} = (\hat{Y}_1, \ldots, \hat{Y}_{n_{\mathrm{tr}}})$.

### 2.1. Baseline: Fairness-Aware Learning via Sensitive Attribute Resampling

We begin by presenting the baseline model developed by Romano et al. (2020), *Fair Dummies Learning (FDL)*, whose high-level model architecture is the same as FairICP. We then discuss the challenge it may face when dealing with complex sensitive attributes.

The key idea of FDL is to construct a synthetic version of the original sensitive attribute as $\tilde{\mathbf{A}}$ based on *conditional randomization* (Candès et al., 2018), drawing independent samples $\tilde{A}_i$ from $Q(\cdot|Y_i)$ for $i = 1, \ldots, n_{tr}$ where the $Q(\cdot|y)$ is the conditional distribution of $A$ given $Y = y$. Since the re-sampled $\tilde{\mathbf{A}}$ is generated independently without looking at the features $\mathbf{X}$, and consequently, the predicted responses $\hat{\mathbf{Y}}$, $\tilde{\mathbf{A}}$ satisfies equalized odds: $\hat{Y} \perp\!\!\!\perp \tilde{A} \mid Y$. Given the resampled sensitive attribute, FDL uses the fact that $A$ satisfies equalized odds if and only if $(\hat{Y}, A, Y) \overset{d}{=} (\hat{Y}, \tilde{A}, Y)$, and promotes equalized odds by enforcing the similarity between joint distributions of $(\hat{Y}, A, Y)$ and $(\hat{Y}, \tilde{A}, Y)$ via an adversarial learning component (Goodfellow et al., 2014), where the model iteratively learn how to separate these two distributions and optimize a fairness-regularized prediction loss. More specifically, define the negative log-likelihood loss, the discriminator loss, and the value function respec-

tively:

$$\mathcal{L}_f(\theta_f) = \mathbb{E}_{XY}\left[-\log p_{\theta_f}(Y \mid X)\right], \tag{2}$$

$$\mathcal{L}_d(\theta_f, \theta_d) = \mathbb{E}_{\hat{Y}AY}[-\log D_{\theta_d}(\hat{Y}, A, Y)]$$
$$+ \mathbb{E}_{\hat{Y}\tilde{A}Y}[-\log(1 - D_{\theta_d}(\hat{Y}, \tilde{A}, Y))], \tag{3}$$

$$V_\mu(\theta_f, \theta_d) = (1 - \mu)\mathcal{L}_f(\theta_f) - \mu\mathcal{L}_d(\theta_f, \theta_d), \tag{4}$$

where $D_{\theta_d}(.)$ is the classifier parameterized by $\theta_d$ which separates the distribution of $(\hat{Y}, A, Y)$ and the distribution of $(\hat{\mathbf{Y}}, \tilde{\mathbf{A}}, \mathbf{Y})$, and $\mu \in [0, 1]$ is a tuning parameter that controls the prediction-fairness trade-off. Then, FDL learns $\theta_f, \theta_d$ by finding the minimax solution

$$\hat{\theta}_f, \hat{\theta}_d = \arg\min_{\theta_f} \max_{\theta_d} V_\mu(\theta_f, \theta_d). \tag{5}$$

**Challenges with Complex Sensitive Attributes**  FDL generates $\tilde{A}$ through conditional randomization and resamples it from the (estimated) conditional distribution $Q(A \mid Y)$. However, FDL was proposed primarily for the scenario with a single continuous sensitive attribute, as the estimation of $Q(A \mid Y)$ is challenging when the dimension of $A$ increases due to the curse of dimensionality (Scott, 2015). For example, with categorical variables, combining categories to model dependencies leads to an exponentially decreasing amount of data in each category, making estimation unreliable. Also, when $A$ includes mixed-type variables, modeling the joint conditional distribution $q(A|Y)$ becomes complex. Therefore, an approach that allows $A$ to have flexible types and scales well with its dimensionality is crucial for promoting improved equalized odds in many social and medical applications.

### 2.2. FairICP: Fairness-Aware Learning via Inverse Conditional Permutation

To circumvent the challenge in learning the conditional density of $A$ given $Y$, we propose the *Inverse Conditional Permutation (ICP)* sampling scheme, which leverages *Conditional Permutation (CP)* (Berrett et al., 2020) but pivots to estimate $Y$ given $A$, to generate a permuted version of $\tilde{\mathbf{A}}$ which is guaranteed to satisfy $(\hat{\mathbf{Y}}, \mathbf{A}, \mathbf{Y}) \overset{d}{=} (\hat{\mathbf{Y}}, \tilde{\mathbf{A}}, \mathbf{Y})$ when the equalized odds defined in eq. (1) holds.

**Recap of CP and Why It's Not Sufficient**  FDL constructs synthetic and resampled sensitive attributes based on conditional randomization. CP offers a natural alternative approach to constructing the synthetic sensitive attribute $\tilde{\mathbf{A}}$ (Berrett et al., 2020). Here, we provide a high-level recap of the CP sampling and demonstrate how we can apply it to generate synthetic sensitive attributes $\tilde{\mathbf{A}}$. Let $\mathcal{S}_n$ denote the set of permutations on the indices $\{1, \ldots, n\}$. Given any vector $\mathbf{x} = (x_1, \ldots, x_n)$ and any permutation $\pi \in \mathcal{S}_n$, define $\mathbf{x}_\pi = (x_{\pi(1)}, \ldots, x_{\pi(n)})$ as the permuted version of $\mathbf{x}$ with its entries reordered according to the permutation $\pi$. Instead of drawing a permutation $\Pi$ uniformly at random, CP assigns unequal sampling probability to permutations based on the conditional probability of observing $A_\Pi$ given

$Y$:

$$\mathbb{P}\{\Pi = \pi \mid \mathbf{A}, \mathbf{Y}\} = \frac{q^n\left(\mathbf{A}_\pi \mid \mathbf{Y}\right)}{\sum_{\pi' \in \mathcal{S}_n} q^n\left(\mathbf{A}_{\pi'} \mid \mathbf{Y}\right)}. \quad (6)$$

Here, $q(\cdot \mid y)$ is the density of the distribution $Q(\cdot \mid y)$ (i.e., $q(\cdot \mid y)$ is the conditional density of $A$ given $Y = y$). We write $q^n(\cdot \mid \mathbf{Y}) := \prod_{i=1}^{n} q(\cdot \mid Y_i)$ to denote the product density. This leads to the synthetic $\tilde{\mathbf{A}} = \mathbf{A}_\Pi$, which, intuitively, could achieve a similar purpose as the ones from conditional randomization for encouraging equalized odds when utilized in constructing the loss eq. (5).

Compared to the conditional randomization strategy in FDL, one strength of CP is that its generated synthetic sensitive attribute $\tilde{\mathbf{A}}$ is guaranteed to retain the marginal distribution of the actual sensitive attribute $A$ regardless of the estimation quality of $q(\cdot|y)$. However, it still relies strongly on the estimation of $q(\cdot|y)$ for its permutation quality and, thus, does not fully alleviate the issue arising from multivariate density estimation as we mentioned earlier.

**ICP Circumvents Density Estimation of $A \mid Y$**  To circumvent this challenge in estimating the multi-dimensional conditional density $q(\cdot|y)$ which can be further complicated by mixed sensitive attribute types, we propose the indirect ICP sampling strategy. ICP scales better with the dimensionality of $A$ and adapts easily to various data types.

ICP begins with the observation that the distribution of $(\mathbf{A}_\Pi, \mathbf{Y})$ is identical as the distribution of $(\mathbf{A}, \mathbf{Y}_{\Pi^{-1}})$. Hence, instead of determining $\Pi$ based on the conditional law of $A$ given $Y$, we first consider the conditional permutation of $Y$ given $A$, which can be estimated conveniently using standard or generalized regression techniques, as $Y$ is typically one-dimensional. We then generate $\Pi$ by applying an inverse operator to the distribution of these permutations. Specifically, we generate $\tilde{\mathbf{A}} = \mathbf{A}_\Pi$ according to the following probabilities:

$$\mathbb{P}\{\Pi = \pi \mid \mathbf{A}, \mathbf{Y}\} = \frac{q^n\left(\mathbf{Y}_{\pi^{-1}} \mid \mathbf{A}\right)}{\sum_{\pi' \in \mathcal{S}_n} q^n\left(\mathbf{Y}_{\pi'^{-1}} \mid \mathbf{A}\right)}. \quad (7)$$

We adapt the *parallelized pairwise sampler* developed for the vanilla CP to efficiently generate ICP samples (see Appendix C), and show that ICP generate valid conditional permutations of $\mathbf{A}$ given any set of its observed value set $\mathbf{a}$.

**Theorem 2.1.** *Let $(\mathbf{X}, \mathbf{A}, \mathbf{Y})$ be i.i.d observations of sample size $n$, $S(\mathbf{A})$ denote the unordered set of rows in $\mathbf{A}$, and $p$ be the dimension of $A$. Let $\tilde{\mathbf{A}}$ be sampled via ICP based on eq. (7). Then,*

*(1) $\tilde{\mathbf{A}}$ is a valid conditional permutation of $\mathbf{A}$: for any $\pi$,*

$$\mathbb{P}\{\mathbf{A} = \mathbf{a}_\pi \mid S(\mathbf{A}) = S, \mathbf{Y}\} = \mathbb{P}\left\{\tilde{\mathbf{A}} = \mathbf{a}_\pi \mid S(\mathbf{A}) = S, \mathbf{Y}\right\}.$$

*(2) If $\hat{Y} \perp\!\!\!\perp A \mid Y$, we have $(\hat{\mathbf{Y}}, \mathbf{A}, \mathbf{Y}) \overset{d}{=} (\hat{\mathbf{Y}}, \tilde{\mathbf{A}}, \mathbf{Y})$.*

Theorem 2.1 is derived from Bayes' Rule, with its proof provided in Appendix B

---

**Algorithm 1** FairICP: Fairness-aware learning via inverse conditional permutation

---

**Input**: Data $(\mathbf{X}, \mathbf{A}, \mathbf{Y}) = \{(X_i, A_i, Y_i)\}_{i \in \mathcal{I}_{\mathrm{tr}}}$
**Parameters**: penalty weight $\mu$, step size $\alpha$, number of gradient steps $N_g$, and iterations $T$.
**Output**: predictive model $\hat{f}_{\hat{\theta}_f}(\cdot)$ and discriminator $\hat{D}_{\hat{\theta}_d}(\cdot)$.

1: **for** $t = 1, \ldots, T$ **do**
2:     Generate permuted copy $\tilde{\mathbf{A}}$ by eq. (7) using ICP as implemented in Appendix C.
3:     Update the discriminator parameters $\theta_d$ by repeating the following for $N_g$ gradient steps:

$$\theta_d \leftarrow \theta_d - \alpha \nabla_{\theta_d} \hat{\mathcal{L}}_d(\theta_f, \theta_d).$$

4:     Update the predictive model parameters $\theta_f$ by repeating the following for $N_g$ gradient steps:

$$\theta_f \leftarrow \theta_f - \alpha \nabla_{\theta_f} \left[(1 - \mu)\hat{\mathcal{L}}_f(\theta_f) - \mu \hat{\mathcal{L}}_d(\theta_f, \theta_d)\right].$$

5: **end for**
**Output**: Predictive model $\hat{f}_{\hat{\theta}_f}(\cdot)$.

---

**FairICP Encourages Equalized Odds with Complex Sensitive Attributes**  We propose *FairICP*, an adversarial learning procedure following the same formulation of the loss function shown previously in the discussion for FDL (Section 2.1) but utilizing the permuted sensitive attributes $\tilde{A}$ using ICP, i.e., eq. (7) which requires estimated $q(y|A)$, as opposed to the one from direct resampling using estimated $q(A|y)$. Let $\hat{\mathcal{L}}_f(\theta_f)$ and $\hat{\mathcal{L}}_d(\theta_f, \theta_d)$ be the empirical realizations of the losses $\mathcal{L}_f(\theta_f)$ and $\mathcal{L}_d(\theta_f, \theta_d)$ defined in (2) and (3) respectively. Algorithm 1 presents the details.

In practice, a fair predictor in terms of *equalized odds* that can simultaneously minimize the prediction loss may not exist (Tang & Zhang, 2022), and the minimizer/maximizer to $L_f(\theta_f)/L_d(\theta_f, \theta_d)$ may not be shared as a result. In this situation, setting $\mu$ to a large value will preferably enforce $f$ to satisfy *equalized odds* while setting $\mu$ close to 0 will push $f$ to purely focus on the prediction loss: an increase in accuracy would often be accompanied by a decrease in fairness and vice-versa.

**ICP Enables Equalized Odds Testing with Complex Sensitive Attributes**  As a by-product of ICP, we can now also conduct more reliable testing of equalized odds violation given complex sensitive attributes. Following the testing procedure proposed in *Holdout Randomization Test* (Tansey et al., 2022) and adopted by Romano et al. (2020) which uses a resampled version of $\tilde{A}$ from the conditional distribution of $A|Y$, we can utilize the conditionally permuted copies to test if equalized odds are violated after model training. Algorithm 2 provides the detailed implementation

of this hypothesis testing procedure: we repeatedly generate synthetic copies $\tilde{\mathbf{A}}$ via ICP and compare $T(\hat{\mathbf{Y}}, \mathbf{A}, \mathbf{Y})$ to those using the synthetic sensitive attributes $T(\hat{\mathbf{Y}}, \tilde{\mathbf{A}}, \mathbf{Y})$ for some suitable test statistic $T$. According to Theorem 2.1, $(\hat{\mathbf{Y}}, \tilde{\mathbf{A}}, \mathbf{Y})$ will have the same distribution as $(\hat{\mathbf{Y}}, \mathbf{A}, \mathbf{Y})$ if the prediction $\hat{Y}$ satisfies equalized odds, consequently, the constructed $p$-values from comparing $T(\hat{\mathbf{Y}}, \mathbf{A}, \mathbf{Y})$ and $T(\hat{\mathbf{Y}}, \tilde{\mathbf{A}}, \mathbf{Y})$ are valid for controlling type-I errors.

**Proposition 2.2.** *Suppose the test observations* $(\mathbf{X}^{te}, \mathbf{A}^{te}, \mathbf{Y}^{te}) = \{(X_i, Y_i, A_i) \text{ for } 1 \leq i \leq n_{\text{te}}\}$ *are i.i.d. and* $\hat{\mathbf{Y}}^{te} = \{\hat{f}(X_i) \text{ for } 1 \leq i \leq n_{\text{te}}\}$ *for a learned model* $\hat{f}$ *independent of the test data. If* $H_0 : \hat{\mathbf{Y}}^{te} \perp\!\!\!\perp \mathbf{A}^{te} \mid \mathbf{Y}^{te}$ *holds, then the output p-value* $p_v$ *of Algorithm 2 is valid, satisfying* $\mathbb{P}\{p_v \leq \alpha\} \leq \alpha$ *for any desired Type I error rate* $\alpha \in [0, 1]$.

---

**Algorithm 2** Hypothesis Test for Equalized Odds with ICP

**Input**: Data $(\mathbf{X}^{te}, \mathbf{A}^{te}, \mathbf{Y}^{te}) = \{(\hat{Y}_i, A_i, Y_i)\}$, $1 \leq i \leq n_{\text{test}}$

**Parameter**: the number of synthetic copies $K$.

1: Compute the test statistic $T$ on the test set: $t^* = T(\hat{\mathbf{Y}}^{te}, \mathbf{A}^{te}, \mathbf{Y}^{te})$.
2: **for** $k = 1, \ldots, K$ **do**
3:   Generate permuted copy $\tilde{\mathbf{A}}_k$ of $\mathbf{A}^{te}$ using ICP.
4:   Compute the test statistic $T$ using fake copy on the test set: $t^{(k)} = T(\hat{\mathbf{Y}}^{te}, \tilde{\mathbf{A}}_k, \mathbf{Y}^{te})$.
5: **end for**
6: Compute the $p$-value:
$$p_v = \frac{1}{K+1}\left(1 + \sum_{k=1}^{K} \mathbb{I}\left[t^* \geq t^{(k)}\right]\right).$$

**Output**: A $p$-value $p_v$ for the hypothesis that *equalized odds* (1) holds.

---

### 2.3. Density Estimation

The estimation of conditional densities is a crucial part of both our method and previous work (Berrett et al., 2020; Romano et al., 2020; Mary et al., 2019; Louppe et al., 2017). However, unlike the previous work which requires the estimation of $A \mid Y$, our proposal looks into the easier inverse relationship of $Y \mid A$. To provide more theoretical insights into how the quality of density estimation affects ICP and CP differently, we have additional analysis in Appendix D.

In practice, ICP can easily leverage the state-of-the-art density estimator and is less disturbed by the increased complexity in $A$, due to either dimension or data types. Unless otherwise specified, in this manuscript, we applied *Masked Autoregressive Flow (MAF)* (Papamakarios et al., 2017) to estimate the conditional density of $Y|A$ when $Y$ is continuous and $A_1, \ldots, A_k$ can take arbitrary data types (dis-

crete or continuous) [1]. In a classification scenario when $Y \in \{0, 1, \ldots, L\}$, one can always fit a classifier to model $Y|A$. To this end, FairICP is more feasible to handle complex sensitive attributes and is suitable for both regression and classification tasks.

## 3. Experiments

In this section, we conduct numerical experiments to examine the effectiveness of the proposed method on both synthetic datasets and real-world datasets. All the implementation details are included in Appendix E.

### 3.1. Simulation Studies

In this section, we conduct simulation studies to (1) assess the quality of the conditional permutations generated by ICP and (2) understand how FairICP leverages these permutations to achieve a more favorable accuracy–fairness trade-off for complex sensitive attributes. For the second task, we run a series of ablation studies, replacing ICP with alternative strategies for generating the "fake copies" $\tilde{A}$. Specifically, we compare FairICP to FairCP—which is an intermediate new procedure that generates $\tilde{A}$ by CP—and FDL (Romano et al., 2020), the previously proposed equalized-odds learning model that uses conditional randomization to generate $\tilde{A}$. All methods use the same model architectures and training schemes for consistency.

3.1.1. THE QUALITY OF CONDITIONAL PERMUTATIONS

First, we investigate whether ICP can generate better conditional permutations than the vanilla CP by comparing them to the oracle permutations (generated using the ground truth in the simulation setting). We measure the *Total Variation* (TV) distance between the distributions of permutations generated by ICP/CP and those of the ground truth on a restricted subset of permutations from swapping operation.

**Simulation Setup.** We generate data as the following: 1) Let $A = (U_1, \ldots, U_{K_0}, U_{K_0+1}, \ldots, U_{K_0+K})\Theta^{1/2}$, where $U_j$ are independently generated from a mixed Gamma distribution $\frac{1}{2}\Gamma(1, 1) + \frac{1}{2}\Gamma(1, 10)$, and $\Theta$ is a randomly generated covariance matrix with $\Theta^{\frac{1}{2}}$ eigenvalues equal-spaced in $[1, 5]$; 2) Generate $Y \sim \mathcal{N}\left(\sqrt{\omega}\sum_{j=1}^{K_0} A_j, \sigma^2 + (1-\omega) * K_0\right)$. Here, $Y$ is influenced only by the first $K_0$ components of $A$, and is independent of the remaining $K$ components. The parameter $\omega \in [0, 1]$ controls the dependence on $A$.

We set $K_0 \in \{1, 5, 10\}$, $K \in \{0, 5, 10, 20, 50, 100\}, \omega = 0.6$, and the sample size for density estimation and quality evaluation are both set to be 200. Since the ground truth dependence structure between the mean of $A$ and $Y$ is linear,

---

[1] In Papamakarios et al. (2017), to estimate $p(U = u \mid V = v)$, $U$ is assumed to be continuous while $V$ can take arbitrary form, but there are no requirements about the dimensionality of $U$ and $V$

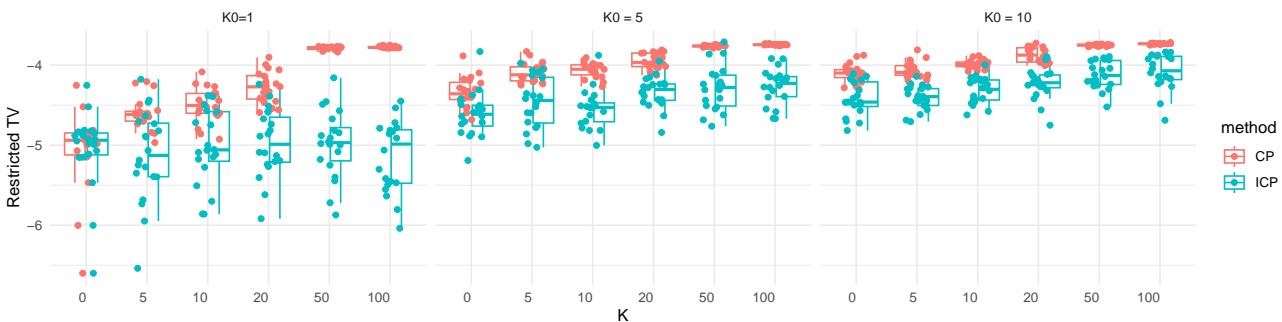

Figure 2: Restricted TV distances ($\log 10$ transformed) between permutations generated by ICP/CP using estimated densities and the oracle permutations generated by true density. Each graph contains results over 20 independent trials as the noise level $K$ increases, with $K_0 = 1, 5, 10$ respectively.

we consider density estimation $\hat{q}_{Y|A}$ based on regularized linear fit when comparing CP and ICP, where we assume $q(y|A)$ or $q(A|y)$ to be Gaussian. We estimate the conditional mean for ICP using LASSO regression (or OLS when $K_0 = 1$ and $K = 0$) with conditional variance based on empirical residuals, and we estimate $q_{A|Y}$ for CP via graphical LASSO (or using empirical covariance when $K_0 = 1$ and $K = 0$). We compare permutations generated by ICP/CP using estimated densities and those using the true density, which is known in simulation up to a normalization constant. Comparisons using the default MAF density estimation are provided in Appendix D.2, which shows the same trend while being uniformly worse for both CP and ICP.

**Evaluation on Permutations Quality.** Due to the large permutation space, the calculation of the actual total variation distance is infeasible. To circumvent this challenge, we consider a restricted TV distance where we restrict the permutation space to swapping actions. Concretely, we consider the TV distance restricted to permutations $\pi$ that swap $i$ and $j$ for $i \neq j, i, j = 1, \ldots, n$ and the original order, and compare ICP/CP to the oracle conditional permutations on such $\frac{n(n-1)}{2}$ permutations only.

**Results** Figure 2 shows restricted TV distance between permutations generated by CP/ICP and the oracle conditional permutations using the true densities, averaged over 20 independent trials. We observe that the restricted TV distances between permutations by ICP and the oracle are much lower compared to those from CP with increased sensitive attribute dimensions, for both dimensions of the relevant sensitive attributes $K_0$ and dimensions of the irrelevant sensitive attributes $K$. These results confirm our expectation that ICP can provide higher-quality sampling by more effectively capturing potentially intrinsic structures between $Y$ and $A$, For instance, when $K_0 = 1$, ICP achieves substantially better estimation quality than CP for moderately large $K$. Additional discussions and mathematical intuitions on why this occurs can be found in Appendix D.1.

### 3.1.2. INFLUENCE OF CP ON FAIRNESS-AWARE LEARNING

Next, we compare the performance of models trained using different resampling methods. Specifically, we compare four models: (1) FairICP (Algorithm 1 with estimated density $\hat{q}(Y|A)$); (3) Oracle (Algorithm 1 with true density $q(Y|A)$); (3) FDL (Romano et al., 2020). Apart from the baseline FDL, we also consider another similar but a new model in our simulation (4) FairCP (Algorithm 1 who are almost identical to FairICP with the only difference being permutations generated by CP using estimated density $\hat{q}(A|Y)$, aiming to investigate if the gain of ICP over CP in generating accurate permutation actually affect the downstream predictive model training. The synthetic experiments allow us to reliably evaluate the violation of the equalized odds using different methods with known ground truth.

**Simulation Setup** We conduct experiments under two simulation settings where $A$ influence $Y$ through $X$, which is the most typical mechanism in the area of fair machine learning (Kusner et al., 2017; Tang & Zhang, 2022; Ghassami et al., 2018).

1. Simulation 1: The response $Y$ depends on two set of features $X^* \in \mathbb{R}^K$ and $X' \in \mathbb{R}^K$:

$$Y \sim \mathcal{N}\left(\Sigma_{k=1}^K X_k^* + \Sigma_{k=1}^K X_k', \sigma^2\right),$$
$$X_{1:K}^* \sim \mathcal{N}(\sqrt{w}A_{1:K}, (1-w)\mathbf{I}_K), X_{1:K}' \sim \mathcal{N}(\mathbf{0}_K, \mathbf{I}_K).$$

2. Simulation 2: The response $Y$ depends on two features $X^* \in \mathbb{R}$ and $X' \in \mathbb{R}$:

$$Y \sim \mathcal{N}\left(X^* + X', \sigma^2\right),$$
$$X^* \sim \mathcal{N}(\sqrt{w}A_1, 1-w), X' \sim \mathcal{N}(0, 1).$$

In both settings, $A$ are generated as in Section 3.1.1: $A = (U_1, \ldots, U_k)\Theta^{1/2}$, where $k = 1, \ldots, K$ for Simulation 1 (where all the $A_{1:K}$ affects $Y$) and $k = 1, \ldots, K + 1$ for Simulation 2 (where only $A_1$ affects $Y$, with the rest serving as noises to increase the difficulty of density estimation). We set $K \in \{1, 5, 10\}, \omega \in \{0.6, 0.9\}$ to investigate different levels of dependence on $A$, and the sample size for training/test data to be 500/400. For all models, we implement the predictor $f$ as linear model and discriminator $d$

as neural networks; for density estimation part, we utilize MAF (Papamakarios et al., 2017) for all methods except the oracle (which uses the true density).

**Evaluation on the Accuracy-Fairness Tradeoff.** For evaluating equalized odds, we use the empirical *KPC* $= \hat{\rho}^2(\hat{Y}, A \mid Y) \in [0,1]$, which is a flexible conditional independence measure allowing different shapes of $A$ and serves as a natural metric for quantifying equalized-odds violations. We also assess whether KPC is a suitable model-comparison metric by examining trade-off curves in terms of (KPC, prediction loss) versus (fairness testing power, prediction loss). The fairness test power is defined as the ability to reject the hypothesis test outlined by Algorithm 2—which uses the true conditional density of $Y|A$ and the KPC statistic $T$ with targeted type I error at $\alpha = 0.05$. The greater $\hat{\rho}^2$ or rejection power indicates stronger conditional dependence between $A$ and $\hat{Y}$ given $Y$. The power metric provides a fair comparison of equalized-odds violations when the true density is known, however, the true density is unavailable in practice, which discounts our trust of it. If the trade-off curves based on KPC closely mirror those based on the power metric under a known density, then KPC can be considered a viable metric for evaluating fairness in real data settings.[1]

**Results** Figure 3 shows the trade-off curves between prediction loss and equalized odds violation measured by KPC and the associated power using Algorithm 2 ($T = KPC$) under the high-dependence scenarios $w = 0.9$ in Simulation 1 and Simulation 2 respectively, with $K \in \{1, 5, 10\}$. We train the predictor $f$ as linear models and the discriminator $d$ as neural networks with different penalty parameters $\mu \in [0, 1]$. The results are based on 100 independent runs with a sample size of 500 for the training set and 400 for the test set. Results from the low-dependence scenarios are provided in Appendix E.1, which convey the same stories.

Figure 3A shows the results from Simulation 1. All approaches reduce to training a plain regression model for prediction when $\mu = 0$, resulting in low prediction loss but a severe violation of fairness (evidenced by large KPC and statistical power); as $\mu$ increases, models pay more attention to fairness (lower KPC and power) by sacrificing more the prediction performance (higher loss). FairICP performs very closely to the oracle model while outperforming both FDL and FairCP as the dimension of $K$ gets larger, which fits our expectation and follows from the increased difficulty of estimating the conditional density of $A|Y$. Figure 3B shows the results from Simulation 2 and delivers a similar message as Figure 3A.

---

[1]Note that, in Simulation 2 only $A_1$ influences the $Y$, so the test will be based on $\hat{\rho}^2(\hat{Y}, A_1 \mid Y)$ to exclude the effects of noise (though the training is based on $A_{1:K+1}$ for all methods to evaluate the performance under noise).

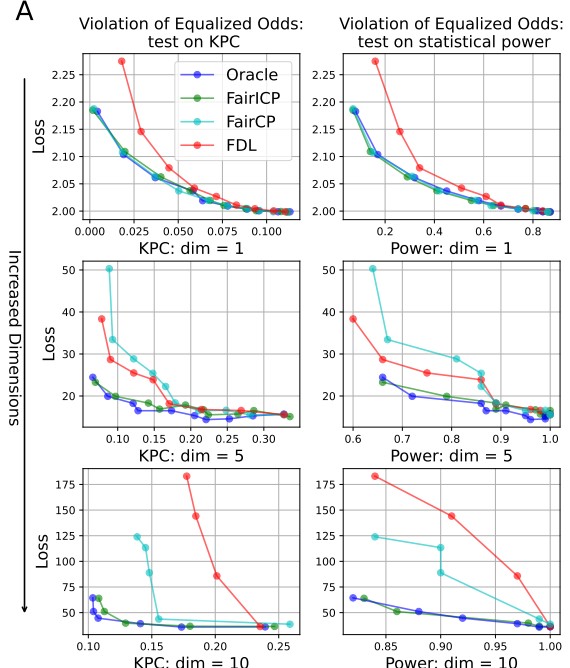

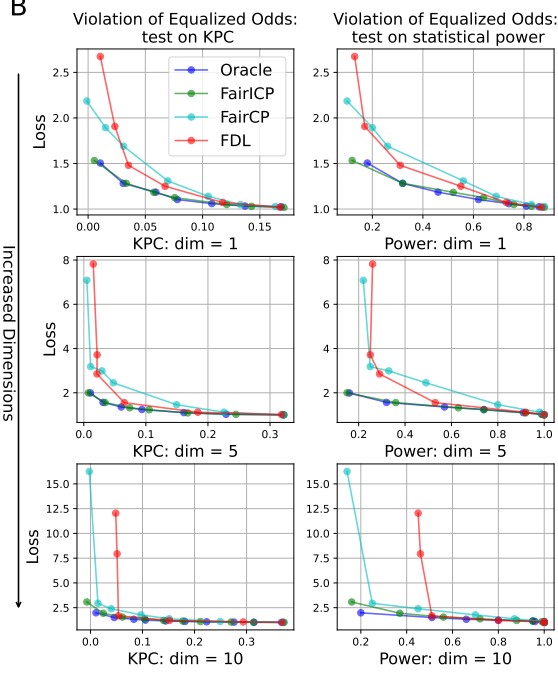

Figure 3: Prediction loss (MSE) and violation of equalized odds in simulation over 100 independent runs under Simulation 1/ Simulation 2 and $w = 0.9$. For each setting, conditional dependence measure KPC and statistical power $\mathbb{P}\{p\text{-value} < 0.05\}$ are shown in the left column and right column respectively. From top to bottom shows the results on different choices of sensitive attribute dimension $K$. The X-axis represents the metrics of equalized odds and the Y-axis is the prediction loss. The Pareto front for each algorithm is obtained by varying the fairness trade-off parameter $\mu$.

| | Crimes (one race) | | Crimes (all races) | | ACS Income | | Adult | | | COMPAS | | |
| --- | --- | --- | --- | --- | --- | --- | --- | --- | --- | --- | --- | --- |
| | Loss (Std) | KPC (Power) | Loss (Std) | KPC (Power) | Loss (Std) | KPC (Power) | Loss (Std) | KPC (Power) | DEO | Loss (Std) | KPC (Power) | DEO |
| Baseline (Unfair) | 0.340(0.039) | 0.130(0.68) | 0.340(0.039) | 0.259(1.00) | 0.210(0.003) | 0.073(1.00) | 0.155(0.004) | 0.042(0.93) | 0.431 | 0.336(0.01) | 0.046(0.41) | 0.858 |
| FairICP | **0.386(0.045)** | **0.016(0.10)** | **0.418(0.047)** | **0.054(0.37)** | **0.215(0.003)** | **0.025(0.80)** | **0.159(0.003)** | **0.010(0.13)** | **0.212** | 0.402(0.02) | 0.008(0.04) | 0.471 |
| HGR | 0.386(0.044) | 0.026(0.16) | 0.596(0.050) | 0.068(0.48) | 0.220(0.004) | 0.021(0.82) | 0.165(0.004) | 0.006(0.10) | 0.198 | 0.443(0.04) | 0.008(0.06) | 0.459 |
| FDL | 0.402(0.046) | 0.023(0.17) | 0.621(0.48) | 0.058(0.37) | / | / | 0.161(0.005) | 0.011(0.26) | 0.251 | 0.417(0.03) | 0.008(0.11) | 0.421 |
| GerryFair | / | / | / | / | 0.262(0.004) | 0.050(1.00) | 0.187(0.004) | 0.031(0.78) | 0.298 | 0.438(0.07) | 0.02(0.14) | 0.368 |
| Reduction | / | / | / | / | / | / | 0.161(0.004) | 0.005(0.15) | 0.212 | **0.400(0.02)** | **0.002(0.05)** | **0.460** |

Table 1: Comparisons of methods encouraging equalized odds across five real data tasks. FairICP (ours), FDL, HGR, and Reduction are compared, with "Baseline (Unfair)" included as a reference which is the pure prediction model. Reported are the mean prediction loss (with standard deviations in parentheses) and the mean KPC for equalized odds violations (with testing power $\mathbb{P}p$-value $< 0.05$ in parentheses). For the *Adult* and *COMPAS* datasets, which use categorical $A$, the DEO equalized odds violation metric is also included. "Fairness trade-off parameters in equalized-odds models are selected to achieve similar violation levels, with the full prediction loss-KPC trade-off curves shown in Figure 4.

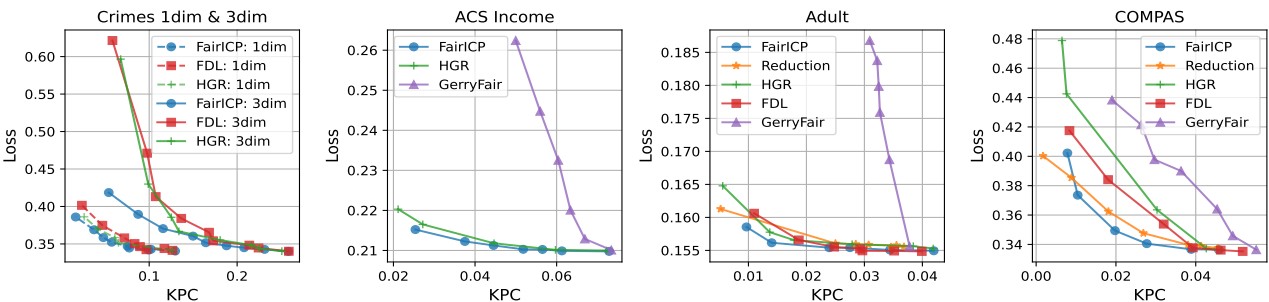

Figure 4: Prediction loss and violation of equalized odds (measured by KPC) obtained by different methods on *Crimes/ACS Income/ Adult/COMPAS* data over 100 random splits. The Pareto front for each algorithm is obtained by varying the fairness trade-off parameter. Similar results measured by testing power is in Appendix E.3.

*Remark* 3.1. The power measure (Algorithm 2) depends on how the permutation/sampling is conducted in practice. In simulations, we can trust it by utilizing the true conditional density, but its reliability hinges on the accuracy of density estimation. In contrast, the direct KPC measure is independent of density estimation.

### 3.2. Experiments on Real Data

In this section, we consider several real-world scenarios where we may need to protect multiple sensitive attributes. For all experiments, the data is repeated divided into a training set (60%) and a test set (40%) 100 times, with the average results on the test sets reported .

- **Communities and Crime Data Set:** This dataset contains 1994 samples and 122 features. The goal is to build a ***regression*** model predicting the continuous violent crime rate. We take the ***continuous percentages of all three minority races*** (African American, Hispanic, Asian, referred to as "3 dim") as sensitive attributes instead of only one race as done in the previous literature. We also consider the case where $A$ only includes one race (African American, referred to as "1 dim") for better comparisons.

- **ACSIncome Dataset:** We use the ACSIncome dataset from Ding et al. (2021) with 100,000 instances (subsampled) and 10 features. The task is a ***binary classification*** to predict if income exceeds $50,000. We consider a

***mixed-type sensitive attributes***: *sex* (male, female), *race* (Black, non-Black), and *age* (continuous).

- **Adult Dataset:** The dataset consists of 48,842 instances and the task is the same as *ACSIncome*. We use both *sex* and *race* as ***two binary sensitive attributes***.

- **COMPAS Dataset:** The ProPublica's *COMPAS* recidivism dataset contains 5278 examples and 11 features (Fabris et al., 2022). The goal is to build a ***binary classifier*** to predict recidivism with ***two binary sensitive attributes*** $A$: *race* (white vs. non-white) and *sex*.

**Results** We compare *FairICP* with four state-of-the-art baselines encouraging equalized odds with the predictor $f$ implemented as a neural network (the results for linear regression/classification is reported in Appendix E.6): *FDL* (Romano et al., 2020), *HGR* (Mary et al., 2019), *GerryFair* (Kearns et al., 2018) and *Exponentiated-gradient reduction* (Agarwal et al., 2018) (referred to as *"Reduction"*). These baselines are originally designed for different tasks. Among them, *Reduction* is designed for binary classification with categorical sensitive attributes (*Adult/COMPAS*), and *Gerry-Fair* adopts a linear threshold method to binarize sensitive attributes for classification (*ACSIncome/Adult/COMPAS*) and targets equal TPR or FPR as an approximation of equalized odds. *HGR* handles both continuous and categorical sensitive attributes for both regression and classification, but how to efficiently generalize it to handle multiple sen-

sitive attributes has not been discussed by the authors [1]. We implement *FDL* with conditional density of $A|Y$ estimated by MAF (Papamakarios et al., 2017) in the continuous case (*Crimes*); for the classification with mixed-type sensitive attributes (*ACSIncome*), it's not straightforward to estimate $A|Y$, so we only consider FDL in the categorical case (*Adult/COMPAS*) where we calculate the frequencies of each class as an estimation of $A|Y$. For our proposed method FairICP, we estimate $Y|A$ with MAF as the same as in FDL in the continuous case, and use a two-layer neural network classifier in the classification (*ACSIncome/Adult/COMPAS*).

In Table 1, we compare FairICP to the baseline methods in terms of predictive performance (MSE for regression and misclassification rate for classification) with model-specific fairness trade-off parameters tuned to yield similar levels of KPC. We find that FairICP achieves the best performance across most tasks, offering lower or similar fairness loss (as measured by KPC or empirical power using estimated density) while also attaining lower prediction loss than the other fairness-aware learning baselines. Although the unfair vanilla baseline achieves the highest prediction accuracy, its equalized-odds violations are several times worse than FairICP's. Finally, FairICP's computational cost is on par with FDL and only slightly higher than HGR (see Appendix E.5 for running time).

Figure 4 shows their full Pareto trade-off curves using KPC (see Appendix E.3 for trade-off curves based on testing powers , Appendix E.4 for trade-off curves based on DEO in *Adult/COMPAS* dataset). These results confirm that the effective multi-dimensional resampling scheme ICP enables FairICP to achieve an improved prediction and equalized odds trade-off compared to existing baselines in the presence of complex and multi-dimensional sensitive attributes.

## 4. Discussion

We introduced a flexible fairness-aware learning approach FairICP to achieve equalized odds with complex sensitive attributes, by combining adversarial learning with a novel inverse conditional permutation strategy. Theoretical insights into the FairICP were provided, and we further conducted numerical experiments on both synthetic and real data to demonstrate its efficacy and flexibility.

Although this work applies ICP within an in-processing framework, the challenge of handling complex sensitive attributes also arises in post-processing approaches. In-processing methods incorporate fairness constraints directly into model training, whereas post-processing adjusts prediction thresholds after training—typically by recalibrating

predicted probabilities across outcome classes (Hardt et al., 2016). Recent work by Tifrea et al. (2023) extends post-processing to handle either a continuous or a categorical sensitive attribute through suitable loss-based optimization. In this context, ICP could serve as a valuable building block to enhance post-processing procedures, particularly by improving the resampling or adjustment steps when working with multi-dimensional $A$.

Finally, we acknowledge the computational overhead associated with adversarial learning, especially on large or complex datasets—a limitation noted in prior work (Zhang et al., 2018; Romano et al., 2020). Future directions include improving the training efficiency of FairICP through stabilization techniques or exploring alternative discrepancy measures such as kernel-based objectives.

## Impact Statement

This work advances the field of fairness-aware machine learning by addressing the underexplored challenge of enforcing equalized odds for multi-dimensional sensitive attributes, such as intersections of race, gender, and socioeconomic status. While our primary contribution is methodological—introducing a theoretically grounded and adaptable framework (FairICP) for multi-attribute fairness—we recognize the broader societal implications of this research. Below, we outline key ethical considerations and potential impacts: 1) By enabling compliance with equalized odds under complex sensitive attributes, FairICP could improve the equity of algorithmic systems in domains like hiring, healthcare, and criminal justice; 2) While our method promotes fairness through adversarial training and inverse conditional permutation, it inherently requires access to sensitive attributes during training. We emphasize that practitioners must carefully evaluate whether collecting such data aligns with ethical and legal standards in their jurisdiction.

In summary, while our work primarily contributes to algorithmic fairness methodology, its societal impact hinges on responsible implementation. We urge practitioners to contextualize FairICP within broader ethical frameworks, engage impacted communities, and prioritize transparency in deployment.

## Acknowledgment

We thank the anonymous reviewers and area chair for their constructive feedback, which helped improve this work. This research was supported in part by NSF grant DMS 2310836.

---

[1]In Mary et al. 2019, since their method can't be directly adapted to multiple sensitive attributes, we compute the mean of the HGR coefficients of each attribute as a penalty.

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

## A. Example of KPC as a Generalized Squared Partial Correlation

We use a simple example to illustrate that KPC can be viewed as the generalized squared partial correlation between $U$ and $V$ given $W$. To see this, we first recall that $MMD^2(P_U, P_V) = \|\mu_U - \mu_V\|^2_{\mathcal{H}}$ where $\mathcal{H}$ is RKHS and $\mu_U = \mathbb{E}k(\cdot, U)$ is the kernel mean embedding of a distribution $P_U$ (Gretton et al., 2012) and consider the special case where the kernel is linear and $U = \alpha W + \beta V + \varepsilon$ with $W, V, \varepsilon$ being independently distributed with mean 0 and variance 1. In this case, we have $P_{U|W,V} = \mathcal{N}(\alpha W + \beta V, 1)$, $P_{U|W} = \mathcal{N}(\alpha W, 1 + \beta^2)$ and $P_U = \mathcal{N}(0, 1 + \alpha^2 + \beta^2)$. Then, the numerator in $KPC(U, V|W)$ becomes: $E[MMD^2(P_{U|W,V}, P_{U|W})] = E[(\alpha W + \beta V - \alpha W)^2] = \beta^2$. The denominator becomes: $E[MMD^2(\delta_U, P_{U|W})] = E[(U - \alpha W)^2] = \beta^2 + 1$. Hence, we can see that $KPC(U, V|W)$ reduces to the squared classical partial correlation between $U$ and $V$ given $W$: $KPC(U, V|W) = (\rho_{UV \cdot W})^2 = \frac{\beta^2}{1 + \beta^2}$. In this special model, we know that $U$ is conditionally independent of $V$ given $W$ if and only if the partial correlation/KPC is 0 ($\beta = 0$).

## B. Proofs

*Proof of Theorem 2.1.* Let $S(\mathbf{A}) = \{A_1, \ldots, A_n\}$ denote the row set of the observed $n$ realizations of sensitive attributes (unordered and duplicates are allowed). Let $\mathbf{X}, \hat{\mathbf{Y}} := f(\mathbf{X})$ and $\mathbf{Y}$ be the associated $n$ feature, prediction, and response observations. Recall that, with a slight abuse of notations, we have used $q(.)$ to denote both the density for continuous variables or potentially point mass for discrete observations. For example, if we have a continuous variable $U$ and discrete variable $V$, then, $q_{U,V}(u, v) = q_{U|V}(u|v)q_V(v)$ with $q_V(v)$ the point mass at $v$ for $V$ and $q_{U|V}(u|v)$ is the conditional density of $U$ given $V = v$. Similar convention is adopted for the definition of $\mathbb{P}$, e.g., $\mathbb{P}(U = u, V = v) = q_{U|V}(u|v)q_V(v)$, $\mathbb{P}(U = u, V \le v) = \sum_{v' \le v} q_{U|V}(u|v)q_V(v')$, $\mathbb{P}(U \le u, V \le v) = \sum_{v' \le v} \int q_{U|V}(u'|v')q_V(v')du'$.

1. Task 1: Show that $\tilde{\mathbf{A}}$ generated by ICP is a valid conditional permutation of $\mathbf{A}$, as generated by CP.

   **Proof of Task 1.** Recall that conditional on $S(\mathbf{A}) = S$ for some $S = \{a_1, \ldots, a_n\}$, we have (Berrett et al., 2020):

$$\mathbb{P}\{\mathbf{A} = \mathbf{a}_\pi | S(\mathbf{A}) = S, \mathbf{Y}\} = \frac{q^n_{A|Y}(\mathbf{a}_\pi \mid \mathbf{Y})}{\sum_{\pi' \in \mathcal{S}_n} q^n_{A|Y}(\mathbf{a}_{\pi'} \mid \mathbf{Y})}, \tag{8}$$

   where $\mathbf{a} = (a_1, \ldots, a_n)$ is the stacked $a$ values in $S$. On the other hand, conditional on $S(\tilde{\mathbf{A}}) = S$, by construction:

$$\mathbb{P}\left\{\tilde{\mathbf{A}} = \mathbf{a}_\pi | S(\mathbf{A}) = S, \mathbf{Y}\right\} = \frac{q^n_{Y|A}(\mathbf{Y}_{\pi^{-1}} \mid \mathbf{a})}{\sum_{\pi'} q^n_{Y|A}(\mathbf{Y}_{\pi'^{-1}}|\mathbf{a})} = \frac{q^n_{A|Y}(\mathbf{a}_\pi \mid \mathbf{Y})}{\sum_{\pi'} q^n_{A|Y}(\mathbf{a}_\pi \mid \mathbf{Y})}. \tag{9}$$

   where the last equality utilizes the following fact,

$$\frac{q^n_{Y|A}(\mathbf{Y}_{\pi^{-1}} \mid \mathbf{a})}{\sum_{\pi'} q^n_{Y|A}(\mathbf{Y}_{\pi'^{-1}}|\mathbf{a})} = \frac{q^n_{Y,A}(\mathbf{Y}_{\pi^{-1}}, \mathbf{a})}{\sum_{\pi' \in \mathcal{S}_n} q^n_{Y,A}(\mathbf{Y}_{\pi'^{-1}}, \mathbf{a})} = \frac{q^n_{Y,A}(\mathbf{Y}, \mathbf{a}_\pi)}{\sum_{\pi' \in \mathcal{S}_n} q^n_{Y,A}(\mathbf{Y}, \mathbf{a}_{\pi'})} = \frac{q^n_{A|Y}(\mathbf{a}_\pi \mid \mathbf{Y})}{\sum_{\pi'} q^n_{A|Y}(\mathbf{a}_{\pi'} \mid \mathbf{Y})}.$$

   Hence, by construction, we must have $\mathbb{P}\left\{\tilde{\mathbf{A}} = \mathbf{a}_\pi | S(\mathbf{A}) = S, \mathbf{Y}\right\} = \mathbb{P}\{\mathbf{A} = \mathbf{a}_\pi | S(\mathbf{A}) = S, \mathbf{Y}\}$

2. Task 2: Show that $(\hat{\mathbf{Y}}, \mathbf{A}, \mathbf{Y}) \stackrel{d}{=} (\hat{\mathbf{Y}}, \tilde{\mathbf{A}}, \mathbf{Y})$ given conditional independence $\hat{Y} \perp\!\!\!\perp A | Y$.

   **Proof of Task 2.** By Task 1, we can show that $\tilde{\mathbf{A}} | \mathbf{Y} \stackrel{d}{=} \mathbf{A} | \mathbf{Y}$:

$$\mathbb{P}(\mathbf{A} \le \mathbf{t} | Y) = \mathbb{E}_{S|\mathbf{Y}}\mathbb{P}(\mathbf{A} \le \mathbf{t} | \mathbf{Y}, S(\mathbf{A}) = S)] = \mathbb{E}_{S|Y}\mathbb{P}(\tilde{\mathbf{A}} \le \mathbf{t} | \mathbf{Y}, S(\mathbf{A}) = S)] = \mathbb{P}(\tilde{\mathbf{A}} \le \mathbf{t} | Y).$$

   Additionally, under the assumption that $A \perp\!\!\!\perp \hat{Y} | Y$, $\tilde{A} \perp\!\!\!\perp \hat{Y} | Y$ by construction since $\tilde{A}$ depends on the observed data only through $Y$ and $S(\mathbf{A})$. Consequently, we have

$$q_{\hat{\mathbf{Y}}, \mathbf{A}, \mathbf{Y}}(\hat{\mathbf{y}}, \mathbf{a}, \mathbf{y}) = q_{\hat{\mathbf{Y}}|\mathbf{Y}}(\hat{\mathbf{y}}|\mathbf{y})q_{\mathbf{A}|\mathbf{Y}}(\mathbf{a}|\mathbf{y})q_{\mathbf{Y}}(\mathbf{y}) = q_{\hat{\mathbf{Y}}|\mathbf{Y}}(\hat{\mathbf{y}}|\mathbf{y})q_{\tilde{\mathbf{A}}|\mathbf{Y}}(\mathbf{a}|\mathbf{y})q_{\mathbf{Y}}(\mathbf{y}) = q_{\hat{\mathbf{Y}}, \tilde{\mathbf{A}}, \mathbf{Y}}(\hat{\mathbf{y}}, \mathbf{a}, \mathbf{y}).$$

$\square$

*Proof of Proposition 2.2.* The proposed test is a special case of the Conditional Permutation Test (Berrett et al., 2020), so the proof is a direct result from Theorem 2.1 in our paper and Theorem 1 in (Berrett et al., 2020) .

$\square$

## C. Sampling Algorithm for ICP

To sample the permutation $\Pi$ from the probabilities:

$$\mathbb{P}\{\Pi = \pi \mid \mathbf{A}, \mathbf{Y}\} = \frac{q^n\left(\mathbf{Y}_{\pi^{-1}} \mid \mathbf{A}\right)}{\sum_{\pi' \in \mathcal{S}_n} q^n\left(\mathbf{Y}_{\pi'^{-1}} \mid \mathbf{A}\right)},$$

we use the *Parallelized pairwise sampler for the CPT* proposed in Berrett et al. (2020), which is detailed as follows:

---

**Algorithm 3** Parallelized pairwise sampler for the ICP

---

**Input**: Data $(\mathbf{A}, \mathbf{Y})$, Initial permutation $\Pi^{[0]}$, integer $S \geq 1$.

1: **for** $s = 1, \ldots, S$ **do**

2:     Sample uniformly without replacement from $\{1, \ldots, n\}$ to obtain disjoint pairs

$$\left(i_{s,1}, j_{s,1}\right), \ldots, \left(i_{s,\lfloor n/2 \rfloor}, j_{s,\lfloor n/2 \rfloor}\right).$$

3:     Draw independent Bernoulli variables $B_{s,1}, \ldots, B_{s,\lfloor n/2 \rfloor}$ with odds ratios

$$\frac{\mathbb{P}\{B_{s,k} = 1\}}{\mathbb{P}\{B_{s,k} = 0\}} = \frac{q\left(Y_{\left(\Pi^{[s-1]}(j_{s,k})\right)} \mid A_{i_{s,k}}\right) \cdot q\left(Y_{\left(\Pi^{[s-1]}(i_{s,k})\right)} \mid A_{j_{s,k}}\right)}{q\left(Y_{\left(\Pi^{[s-1]}(i_{s,k})\right)} \mid A_{i_{s,k}}\right) \cdot q\left(Y_{\left(\Pi^{[s-1]}(j_{s,k})\right)} \mid A_{j_{s,k}}\right)}.$$

    Define $\Pi^{[s]}$ by swapping $\Pi^{[s-1]}\left(i_{s,k}\right)$ and $\Pi^{[s-1]}\left(j_{s,k}\right)$ for each $k$ with $B_{s,k} = 1$.

4: **end for**

**Output**: Permuted copy $\tilde{\mathbf{A}} = \mathbf{A}_{\Pi^{[S]-1}}$.

---

## D. Additional Comparisons of CP/ICP

When we know the true conditional laws $q_{Y|A}(.)$ (conditional density $Y$ given $A$) and $q_{A|Y}(.)$ (conditional density $A$ given $Y$), both CP and ICP show provide accurate conditional permutation copies. However, both densities are estimated in practice, and the estimated densities are denoted as $\check{q}_{Y|A}(.)$ and $\check{q}_{A|Y}(.)$ respectively. The density estimation quality will depend on both the density estimation algorithm and the data distribution. While a deep dive into this aspect, especially from the theoretical aspects, is beyond the scope, we provide some additional heuristic insights to assist our understanding of the potential gain of ICP over CP.

### D.1. When ICP Might Improve over CP?

According to proof argument of Theorem 4 in Berrett et al. (2020), let $\mathbf{A}_{\pi_m}$ be some permuted copies of $A$ according to the estimated conditional law $\check{q}_{A|Y}()$, an upper bound of exchangeability violation for $\mathbf{A}$ and $\mathbf{A}_\pi$ is related to the total variation between the estimated density $\check{q}_{A|Y}(.)$ and $q_{A|Y}(.)$ (Theorem 4 in Berrett et al. (2020)):

$$d_{TV}\{((\mathbf{Y}, \mathbf{A}), (\mathbf{Y}, \mathbf{A}_\pi))|\mathbf{Y}), ((\mathbf{Y}, \check{\mathbf{A}}), (\mathbf{Y}, \mathbf{A}_\pi))|\mathbf{Y})\}$$
$$\leq d_{TV}(\prod_{i=1}^n \check{q}_{A|Y}(.|y_i), \prod_{i=1}^n q_{A|Y}(.|y_i)) \overset{(b_1)}{\leq} \sum_{i=1}^n d_{TV}(\check{q}_{A|Y}(.|y_i), q_{A|Y}(.|y_i)), \qquad (10)$$

where step $(b_1)$ is from Lemma (B.8) from Ghosal & van der Vaart (2017). We adapt the proof arguments of Theorem 4 in Berrett et al. (2020) to the ICP procedure.

Specifically, let $\mathbf{Y}_\pi$ be the conditional permutation of $\mathbf{Y}$ according to $\check{q}_{Y|A}(.)$ and $\check{\mathbf{Y}}$ be a new copy sampled according to $\check{q}_{Y|A}(.)$. We will have

$$d_{TV}\{((\mathbf{Y}, \mathbf{A}), (\mathbf{Y}_\pi, \mathbf{A})|\mathbf{A})\} \leq \sum_{i=1}^n d_{TV}(\check{q}_{Y|A}(.|A_i), q_{Y|A}(.|A_i)). \qquad (11)$$

There is one issue before we can compare the two CP and ICP upper bounds for exchangeability violations: the two bounds consider different variables and conditioning events. Notice that we care only about the distributional level comparisons, hence, we can apply permutation $\pi^{-1}$ to $(\mathbf{Y}, \mathbf{A})$ and $(\mathbf{Y}, \mathbf{A}_{\pi^{-1}})$. The resulting $(\mathbf{Y}_{\pi^{-1}}, \mathbf{A}_{\pi^{-1}})$ is equivalent to $(\mathbf{Y}, \mathbf{A})$ and the resulting $(\mathbf{Y}, \mathbf{A}_{\pi^{-1}})$ is exactly the ICP conditionally permuted version. Next we can remove the conditioning event by marginalizing out $\mathbf{Y}$ and $\mathbf{A}$ in (10) and (11) respectively. Hence, we obtain upper bounds for violation of exchangeability using CP and ICP permutation copies, which is smaller for ICP if $\check{q}_{Y|A}(.)$ is more accurate on average:

$$\mathbb{E}_A\left[d_{TV}(\check{q}_{Y|A}(.|A), q_{Y|A}(.|A))\right] < \mathbb{E}_Y\left[d_{TV}(\check{q}_{A|Y}(.|Y), q_{A|Y}(.|Y))\right].$$

**D.2. ICP Achieved Higher Quality Empirically with MAF Density Estimation**

Here, we compare ICP and CP using MAF-generated densities. The data-generating process is the same as Section 3.1. Note that by design, the linear fit shown in the main paper is favored over MAF for estimating $q_{Y|A}$ in this particular example. There may be better density estimation choices in other applications, but overall, estimating $Y|A$ can be simpler and allows us to utilize existing tools, e.g., those designed for supervised learning.

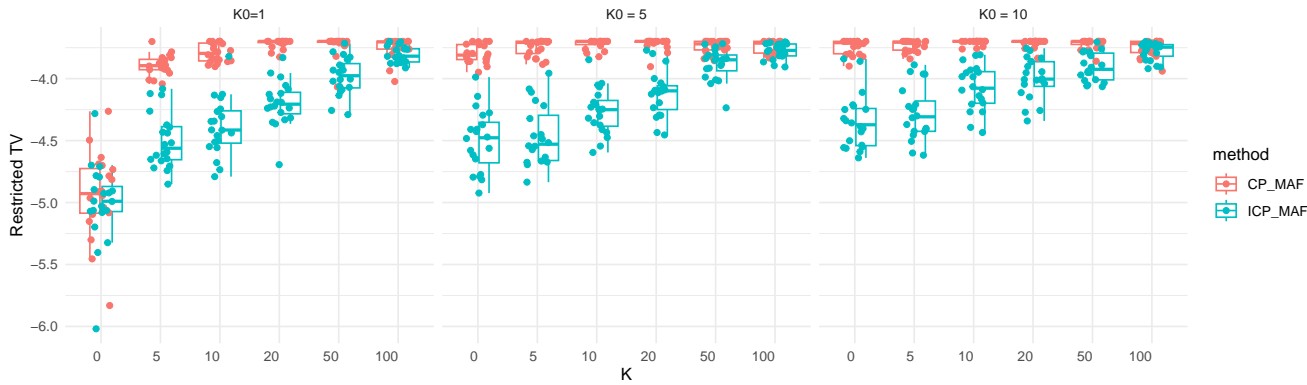

Figure 5: Restricted TV distances (log 10 transformed) between permutations generated by ICP/CP using estimated densities by MAF and the oracle permutations generated by true density. Each graph contains results over 20 independent trials as the noise level $K$ increases, with $K_0 = 1, 5, 10$ respectively.

# E. Experiments on Fairness-Aware Learning Methods Comparisons

In both simulation studies and real-data experiments, we implement the algorithms with the hyperparameters chosen by the tuning procedure as in Romano et al. (2020), where we tune the hyperparameters only once using 10-fold cross-validation on the entire data set and then treat the chosen set as fixed for the rest of the experiments. Then we compare the performance metrics of different algorithms on 100 independent train-test data splits. This same tuning and evaluation scheme is used for all methods, ensuring that the comparisons are meaningful. For KPC (Huang et al., 2022), we use R Package KPC (Huang, 2022) with the default Gaussian kernel and other parameters.

**E.1. Simulation Studies**

For all the models evaluated (FairICP, FairCP, FDL, Oracle), we set the hyperparameters as follows:

- We set $f$ as a linear model and use the Adam optimizer with a mini-batch size in $\{16, 32, 64\}$, learning rate in $\{$1e-4, 1e-3, 1e-2$\}$, and the number of epochs in $\{20, 40, 60, 80, 100, 120, 140, 160, 180, 200\}$. The discriminator is implemented as a four-layer neural network with a hidden layer of size 64 and ReLU non-linearities. We use the Adam optimizer, with a fixed learning rate of 1e-4.

For the MAF used to estimate the conditional density ($Y|A$ and $A|Y$) in the training phase, we use MAF with one MADE component and one hidden layer with $2 *$ conditional inputs nodes, and for optimizer we choose Adam with 0.01 $l_1$ penalty and 0.1 learning rate.

### E.1.1. LOW SENSITIVE ATTRIBUTE DEPENDENCE CASES

We report the results with A-dependence $w = 0.6$ in Figure 6.

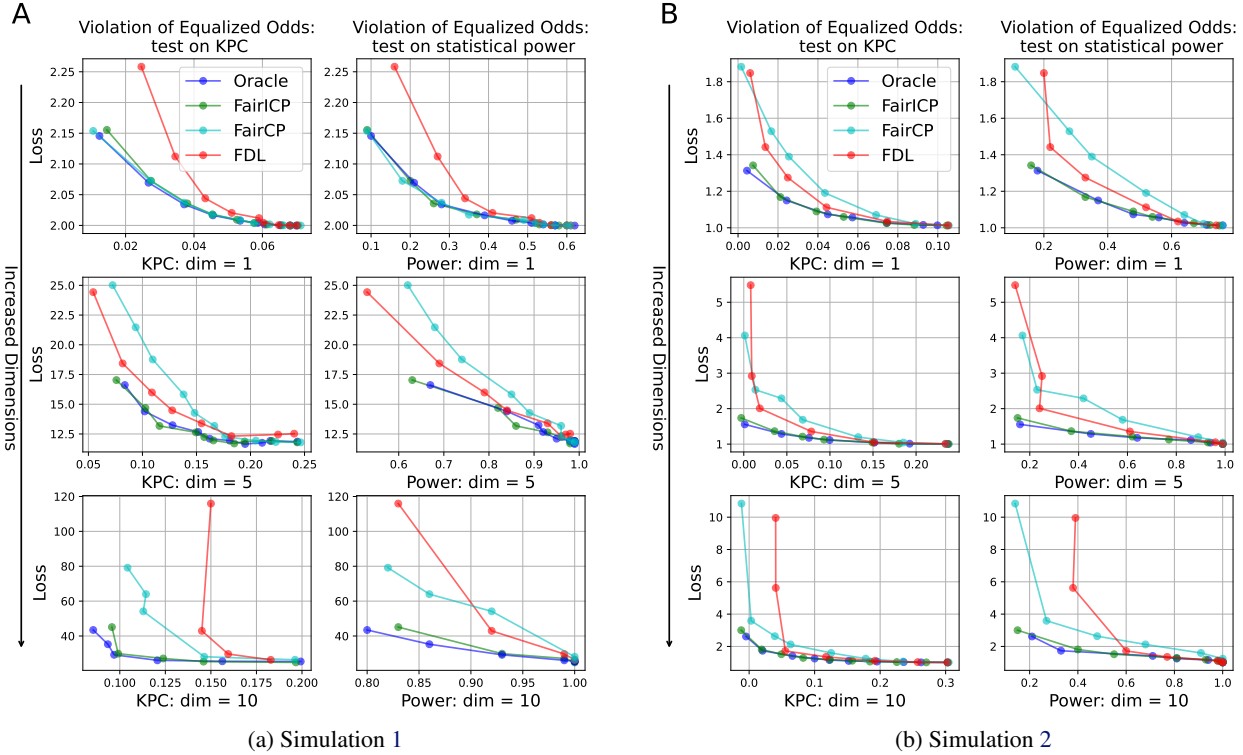

(a) Simulation 1             (b) Simulation 2

Figure 6: Prediction loss and metrics of fairness in simulation over 100 independent runs under Simulation 1/Simulation 2 and $w = 0.6$. For each setting, conditional dependence measure KPC and statistical power $\mathbb{P}\{p\text{-value} < 0.05\}$ are shown in the left column and right column respectively. From top to bottom shows the results on different choices of the noisy sensitive attribute dimension of $K$. The X-axis represents the metrics of fairness and the Y-axis is the prediction loss. Each graph shows the proposed method FairICP, FDL, and oracle model with different hyperparameters $\mu$.

### E.2. Real Data Model Architecture

**Regression Tasks** For FairICP and FDL (code is adapted from `https://github.com/yromano/fair_dummies`), the hyperparameters used for linear model and neural network are as follows:

- Linear: we set $f$ as a linear model and use the Adam optimizer with MSE loss and a mini-batch size in $\{16, 32, 64\}$, learning rate in $\{1e\text{-}4, 1e\text{-}3, 1e\text{-}2\}$, and the number of epochs in $\{20, 40, 60, 80, 100\}$. The discriminator is implemented as a four-layer neural network with a hidden layer of size 64 and ReLU non-linearities. We use the Adam optimizer, with cross entropy loss and a fixed learning rate of 1e-4. The penalty parameter $\mu$ is set as $\{0, 0.2, 0.3, 0.4, 0.5, 0.6, 0.7, 0.8, 0.9\}$.

- Neural network: we set $f$ as a two-layer neural network with a 64-dimensional hidden layer and ReLU activation function. The hyperparameters are the same as the linear ones.

- Density estimation of $Y|A$ and $A|Y$: we use MAF with 5 MADE components and two hidden layers of 64 nodes each. We use Adam optimizer with 0.001 learning rate.

For HGR (code is adapted from `https://github.com/criteo-research/continuous-fairness`), the hyperparameters used for the linear model and neural network are as follows:

- Linear: we set $f$ as a linear model and use the Adam optimizer with MSE loss and a mini-batch size in $\{16, 32, 64\}$, learning rate in $\{1e\text{-}4, 1e\text{-}3, 1e\text{-}2\}$, and the number of epochs in $\{20, 40, 60, 80, 100\}$. The penalty parameter $\lambda$ is set as $\{0, 0.25, 0.5, 0.75, 1, 2, 4, 8, 16\}$.

- Neural network: we set $f$ as a two-layer neural network with a 64-dimensional hidden layer and SeLU activation function. The hyperparameters are the same as the linear ones.

**Classification Tasks**    For FairICP and FDL, the hyperparameters used for linear model and neural network are as follows:

- Linear: we set $f$ as a linear model and use the Adam optimizer with cross entropy loss and a mini-batch size in $\{128, 256\}$, learning rate in $\{1e-4, 1e-3, 1e-2\}$, and the number of epochs in $\{20, 40, 60, 80, 100, 120, 120, 140, 160, 180, 200\}$. The discriminator is implemented as a four-layer neural network with a hidden layer of size 64 and ReLU non-linearities. We use the Adam optimizer, with cross-entropy loss and a fixed learning rate in $\{1e-5, 1e-4, 1e-3\}$. The penalty parameter $\mu$ is set as $\{0, 0.3, 0.5, 0.7, 0.8, 0.9\}$.

- Neural network: we set $f$ as a two-layer neural network with a 64-dimensional hidden layer and ReLU activation function. The hyperparameters are the same as the linear ones.

- Density estimation of $Y|A$: we use a two-layer neural network classifier with 64 hidden nodes and ReLU. We use Adam optimizer with cross-entropy loss and a 0.001 learning rate.

For HGR, the hyperparameters used for the linear model and neural network are as follows:

- Linear: we set $f$ as a linear model and use the Adam optimizer with cross entropy loss and a mini-batch size in $\{64, 128, 256\}$, learning rate in $\{1e-4, 1e-3, 1e-2\}$, and the number of epochs in $\{20, 40, 60, 80, 100\}$. The penalty parameter $\lambda$ is set as $\{0, 0.0375, 0.075, 0.125, 0.25, 0.5, 0.75, 1, 1.5\}$.

- Neural network: we set $f$ as a two-layer neural network with a 64-dimensional hidden layer and SeLU activation function. The hyperparameters are the same as the linear ones.

For GerryFair (code is adapted from `https://github.com/algowatchpenn/GerryFair`), the hyperparameters used for the linear model and neural network are as follows:

- Linear: we use the default linear regression in sklearn. The iteration of fictitious play is 500, and the trade-off parameter is in [0.001, 0.03]. We choose FPR as the fairness metric.

- Neural network: we set $f$ as a two-layer neural network with a 64-dimensional hidden layer and ReLU activation function. Adam optimizer is used with a learning rate set in $\{0.001, 0.005\}$ and batch size in $\{128, 256\}$. The rest of the parameters are the same as in the linear case.

For Reduction (we use the package from `https://github.com/fairlearn/fairlearn`), the hyperparameters used for the linear model and neural network are as follows:

- Linear: we use the default logistic regression in sklearn. The maximum iteration is 50, and the trade-off parameter is in [0.5, 100].

- Neural network: we set $f$ as a two-layer neural network with a 64-dimensional hidden layer and ReLU activation function. Adam optimizer is used with a learning rate set in $\{0.001, 0.0001\}$ and batch size in $\{128, 256\}$. The rest of the parameters are the same as in the linear case.

### E.3. Pareto Trade-Off Curves Based on Equalized Odds Testing Power

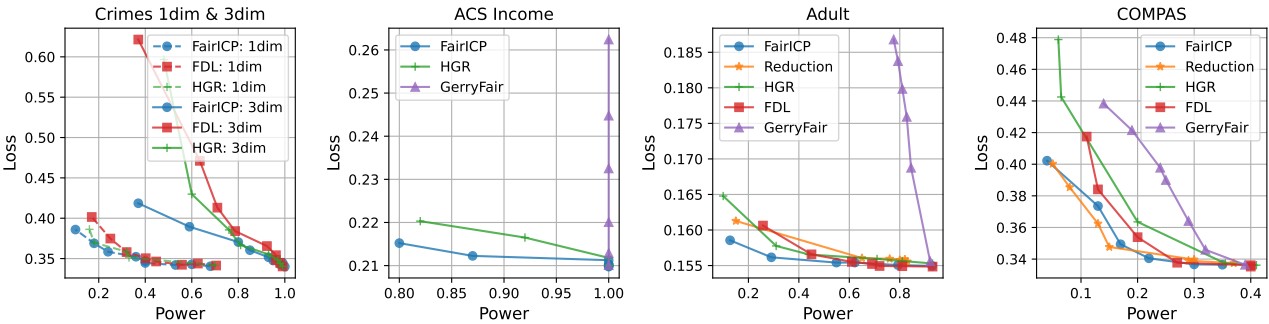

Figure 7: Prediction loss and violation of equalized odds (measured by Power) obtained by different methods on *Crimes/ACS Income/Adult/COMPAS* data over 100 random splits. The Pareto front for each algorithm is obtained by varying the fairness trade-off parameter.

## E.4. Pareto Trade-Off Curves Based on DEO

Apart from KPC and the corresponding testing power, we also consider the standard fairness metric based on the confusion matrix (Hardt et al., 2016; Cho et al., 2020) designed for a binary classification task with categorical sensitive attributes to quantify equalized odds:

$$\text{DEO} := \sum_{y \in \{0,1\}} \sum_{z \in \mathcal{Z}} |\Pr(\hat{Y} = 1 \mid Z = z, Y = y) - \Pr(\hat{Y} = 1 \mid Y = y)|,$$

where $\hat{Y}$ is the predicted class label.

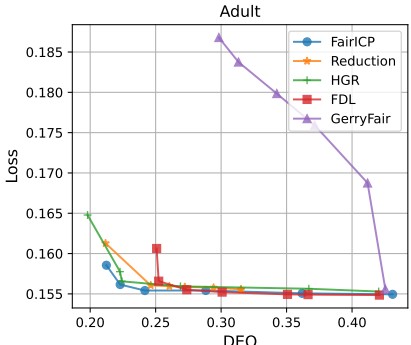 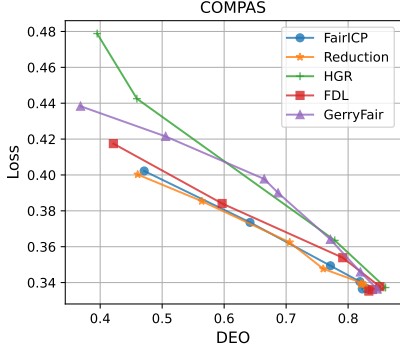

Figure 8: Prediction loss and violation of equalized odds (measured by DEO) obtained by different methods on *Adult*/*COMPAS* data over 100 random splits. The Pareto front for each algorithm is obtained by varying the fairness trade-off parameter.

## E.5. Running Time

We report the running time with neural networks as below:

|            | Crimes (one race) | Crimes (all races) | ACS Income | Adult  | COMPAS |
|------------|-------------------|--------------------|------------|--------|--------|
| FairICP    | 29.4              | 34.6               | 680.7      | 293.1  | 59.8   |
| HGR        | 14.6              | 17.8               | 309.8      | 98.2   | 61.4   |
| FDL        | 28.9              | 39.2               | /          | 289.4  | 67.6   |
| GerryFair  | /                 | /                  | 2834.4     | 1487.4 | 194.8  |
| Reduction  | /                 | /                  | /          | 334.1  | 171.1  |

Table 2: The running time (in seconds) to run a single point on the trade-off curve for each method. Each number is an average of 5 trials.

## E.6. Pareto Trade-Off Curves for Linear Models

We report the results with $f$ as a linear model in Figure 9 for the Communities and Crime dataset (regression), in Figure 11 for the Adult dataset (classification) and in Figure 12 for the COMPAS dataset (classification), which are similar to NN version.

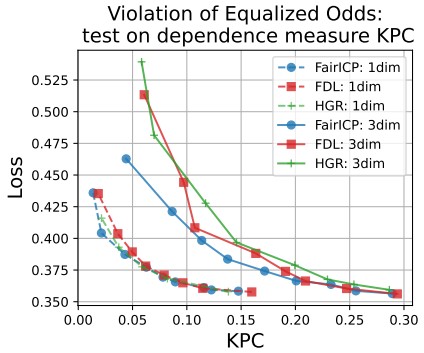 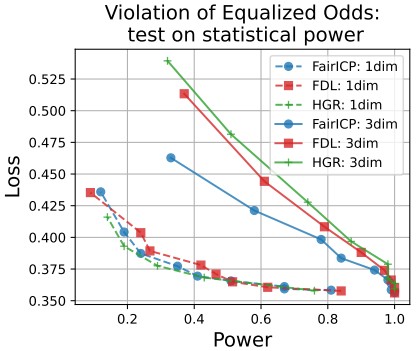

Figure 9: Prediction loss and violation of equalized odds (measured by KPC and statistical power $\mathbb{P}\{p\text{-value} < 0.05\}$) obtained by 3 different training methods in Communities and Crime data over 100 random splits. Each graph shows the results of using different $A$: 1 dim = (African American) and 3 dim = (African American, Hispanic, Asian). The Pareto front for each algorithm is obtained by varying the fairness trade-off parameter.

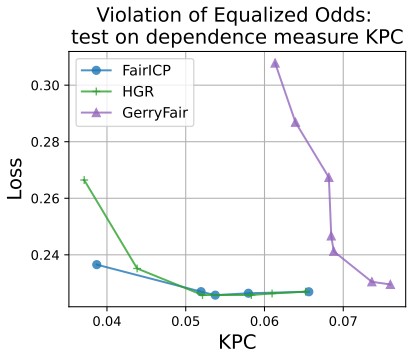 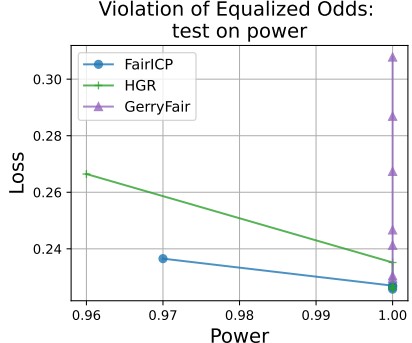

Figure 10: Prediction loss and violation of equalized odds (measured by KPC and statistical power $\mathbb{P}\{p\text{-value} < 0.05\}$) obtained by 3 different training methods in ACS Income data over 100 random splits. The Pareto front for each algorithm is obtained by varying the fairness trade-off parameter.

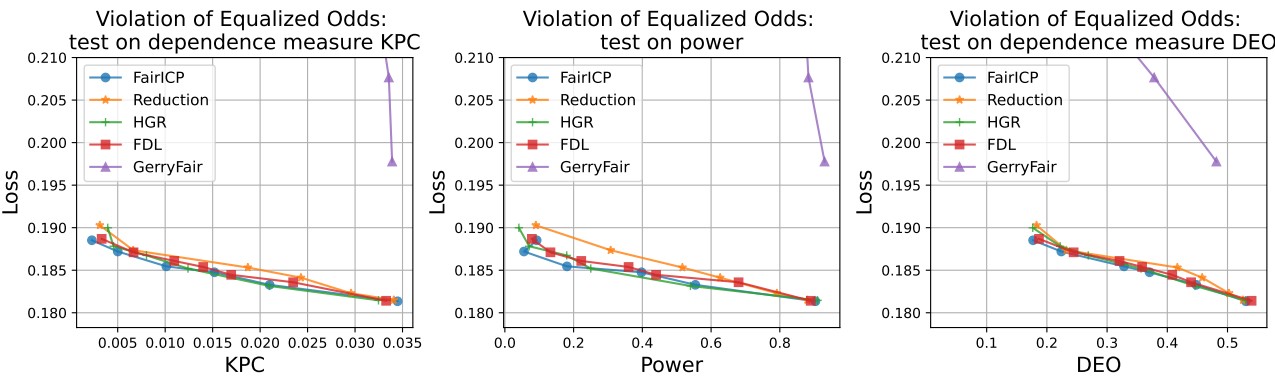

Figure 11: Prediction loss and violation of equalized odds (measured by KPC, statistical power $\mathbb{P}\{p\text{-value} < 0.05\}$ and DEO) obtained by 5 different training methods in Adult data over 100 random splits. The Pareto front for each algorithm is obtained by varying the fairness trade-off parameter. Some points of *GerryFair* are out of the graph on the right.

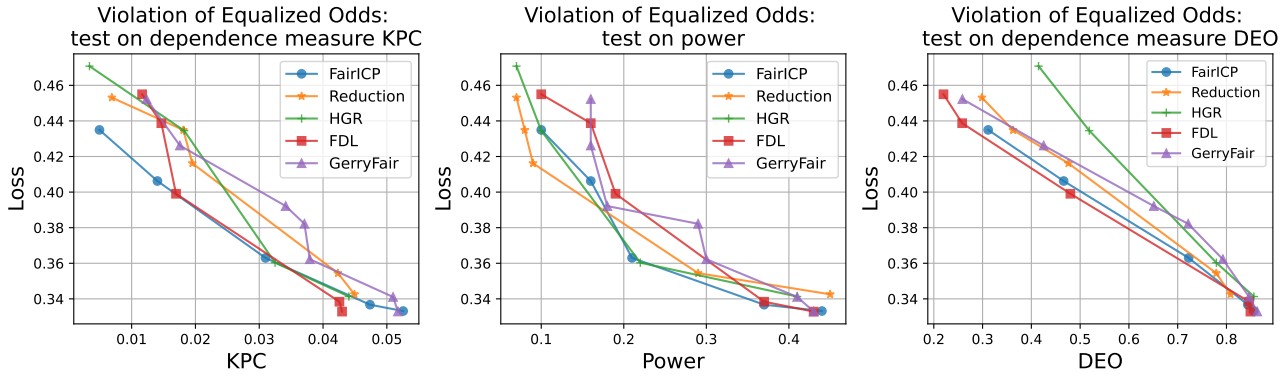

Figure 12: Prediction loss and violation of equalized odds (measured by KPC, statistical power $\mathbb{P}\{p\text{-value} < 0.05\}$ and DEO) obtained by 5 different training methods in COMPAS data over 100 random splits. The Pareto front for each algorithm is obtained by varying the fairness trade-off parameter.

