# OpenReview forum: "FairICP: Encouraging Equalized Odds via Inverse Conditional Permutation"
_ICML.cc/2025/Conference — ICML 2025 poster_

### Official Review · Reviewer_pguj · 2025-03-04

**Overall Recommendation:** 3

**Summary:**

This paper introduces FairICP, a novel fairness-aware learning method designed to promote equalized odds in machine learning models when dealing with complex and multi-dimensional sensitive attributes. The method combines adversarial learning with an innovative Inverse Conditional Permutation (ICP) strategy to generate conditionally permuted copies of sensitive attributes without estimating multi-dimensional conditional densities. The paper provides a comprehensive theoretical foundation for the method and evaluates its performance through extensive simulations and real-world datasets.

## Update after rebuttal

Thanks for your explanation, I will maintain my rating.

**Claims And Evidence:**

Yes, the claims made in the submission are supported by clear and convincing evidence.

**Essential References Not Discussed:**

The paper cites relevant prior work in the field of fairness-aware machine learning. However, it could benefit from additional references to recent advancements in multi sensitive attributes fairness techniques.

**Experimental Designs Or Analyses:**

The experimental designs and analyses are generally sound. The authors conduct extensive experiments on both synthetic and real-world datasets, providing a comprehensive evaluation of the method's performance. The experimental results are presented in a clear and organized manner, and the comparisons with baseline methods are thorough. However, the paper could benefit from a sensitivity analysis to assess the impact of hyperparameters and data characteristics (how complex the sensitive attributes are) on the method's performance.

**Methods And Evaluation Criteria:**

Yes, the proposed methods make sense for the problem (improving fairness measure EO) at hand.

**Other Comments Or Suggestions:**

Providing detailed information on the exact hyper parameter values or the code for reproducing the results would be more convincing.

**Other Strengths And Weaknesses:**

**Strengths:**
1. The introduction of the ICP strategy is a significant contribution that addresses the challenge of handling multi-dimensional sensitive attributes.
2. The integration of ICP with adversarial learning provides a flexible and efficient framework for promoting equalized odds.
3. The empirical validation on both synthetic and real-world datasets demonstrates the effectiveness of the method.

**Weakness:**
1. My concern is that the ICP strategy heavily relies on the accurate estimation of the conditional density $q(Y|A)$. Could you discuss more on the challenges and potential inaccuracies associated with density estimation, especially in high-dimensional spaces or with complex data types.
2. I feel like the experiments are conducted on a limited set of datasets. The number of sensitive attributes is limited. And for the categorical case, multi sensitive attributes can be actually turned to one attribute case by separating the dataset into more subgroups. I am curious about the performance of standard fairness-aware method in this naive way. Maybe you can add this as baseline.
3. While the paper compares FairICP with several baseline methods, it does not include some state-of-the-art methods for fairness-aware learning.
4. The ICP strategy is specifically designed for EO and may not be generalized to other fairness measures enough.

**Questions For Authors:**

Please see weakness.

**Relation To Broader Scientific Literature:**

The key contributions of the paper are well-related to the broader scientific literature. The paper builds upon existing work in fairness-aware machine learning and addresses the underexplored challenge of handling multi-dimensional sensitive attributes. The authors provide a comprehensive review of related work, including prior methods for achieving equalized odds and the limitations of existing approaches.

**Theoretical Claims:**

The theoretical claims in the paper are supported by detailed proofs. The authors provide proofs for the validity of the ICP-generated permutations and their ability to promote equalized odds. The proofs are mathematically sound and well-structured. However, the robustness of these theoretical claims under model misspecification or inaccurate density estimation could be further explored.

---

> ### Author Rebuttal · Authors · 2025-04-01
>
> We appreciate the feedbacks by the reviewer, here are our responses:
>
> 1. "Could you discuss more on the challenges and potential inaccuracies associated with density estimation, especially in high-dimensional spaces or with complex data types."
>
>     **Reply.** Thank you for this question. We mention how the quality of density estimation will affect the performance of our ICP (and CP) sampling scheme in line 241-243, and we also provide a theoretical analysis linked to it in the Appendix C.1. Specifically, we show an smaller TV distance bound for ICP compared with CP when density estimation is inaccurate, which aligns with our claims and experimental evidence (Figure 2).
>
>
> 2. "And for the categorical case, multi sensitive attributes can be actually turned to one attribute case by separating the dataset into more subgroups. I am curious about the performance of standard fairness-aware method in this naive way. Maybe you can add this as baseline."
>
>     **Reply.** We thank the reviewer for this suggestion. We now have also added two more baselines in experiments: FDL [3] and Kearns et al. [2].   [Link for experiments added](https://docs.google.com/document/d/1CLwxWwBRsVrYhhIfeC3_IsDBZ0mAoRiZVWWGLv4DDIw/edit?usp=sharing). We also tried Kearns et al. [2] for Adult dataset but it failed to converge with limited number of iteration for the short time given (as evidenced by Figure 1 in its paper), however, we agree this is an important baseline and are working to add it to our revised paper, and we will also update the results once it is done.
>
> 3. While the paper compares FairICP with several baseline methods, it does not include some state-of-the-art methods for fairness-aware learning.
>
>     **Reply.** Thanks for your feedback. We fully understand the need to compare with other methods. However, based on the literature we have searched so far, what we compared in our paper are already the most recent methods that can be applied to our setting (equalized odds, multiple (mixed) sensitive attributes). We would also appreciate it if the reviewer could point out some other cutting-edge fair machine learning methods suitable to our task. Additionally, the major novelty of this work is the ICP sampling for fairness-aware (equalized odds) learning with multiple sensitive attributes. Hence, we should consider the comparison between FairICP and FDL [3] to be the most informative and reliable evaluation of this paper's contribution in a setting where FDL is straightforward to apply.
>
> 4. The ICP strategy is specifically designed for EO and may not be generalized to other fairness measures enough.
>
>     **Reply.** Thank you for pointing this out. Our focus on the equalized odds metric is intentional, as enforcing equalized odds requires ensuring conditional independence. This conditional nature makes it a more challenging fairness criterion compared to unconditional ones like demographic parity (Tang et al.[1]).
>
>     While ICP is well-suited for enforcing equalized odds due to its ability to avoid the challenging density estimation of given, for unconditional fairness notions such as demographic parity, we can utilize unconditional permutation of which is a special case of FairICP without conditioning.
>
> 5. Providing detailed information on the exact hyper parameter values or the code for reproducing the results would be more convincing.
>
>     **Reply.** We apologize for not including code in the initial submission. We have now provided the relevant code [Link for code](https://drive.google.com/file/d/1gX4UIDPGKYEcL-yK9TtYmazi28gc_4tv/view?usp=sharing).
>
>
> ## Reference
>
> [1] Tang et al. Attainability and optimality: The equalized odds fairness revisited.
>
>
> [2] Kearns et al.. Preventing Fairness Gerrymandering: Auditing and Learning for Subgroup Fairness.
>
> [3] Romano et al. Achieving equalized odds by resampling sensitive attributes.

---

> > ### Comment · Reviewer_pguj · 2025-04-06
> >
> > Thanks for your explanation and I will maintain my score.

---

### Official Review · Reviewer_NoT1 · 2025-03-05

**Overall Recommendation:** 3

**Summary:**

- This study introduces a permutation-based learning algorithm for developing fair predictive models.
- The fairness notion considered in this work is equalized odds.
- The proposed permutation mechanism generates pseudo-sensitive attributes with aligning the distributions of $(\hat{Y}, A, Y)$ and $(\hat{Y}, \tilde{A}, Y)$.
- The proposed generation mechanism is designed to accommodate both continuous and discrete sensitive attributes.
- Experimental results suggest that the proposed method achieves a superior fairness-prediction trade-off compared to two baseline approaches.

## update after rebuttal

- Thanks to the rebuttal, which includes the additional experiments, I will raise my rating to 3.

**Claims And Evidence:**

- Overall, the flexibility of the proposed algorithm in handling various types of sensitive attributes is supported by the experimental results.
- However, while the authors claim that the proposed method offers a superior fairness-accuracy trade-off, it seems not consistently outperforms the baselines when the prediction model is linear.

**Essential References Not Discussed:**

- Please see `Baseline methods’ in **Methods And Evaluation Criteria**.

**Experimental Designs Or Analyses:**

- Implementation details
   - How stable is the adversarial learning process? Given that stability is critical for practical deployment, analyzing it would be valuable.

**Methods And Evaluation Criteria:**

- Metrics:
   - (1) The authors use "Loss" as the primary measure of prediction performance. Including accuracy as an additional metric in classification task would enrich the evaluation part.
   - (2) KPC: The paper primarily employs KPC as the fairness metric. Could the authors clarify why standard measures such as TPR, FPR, or DEO were not mainly used instead?
   - (3) DEO: The paper defines DEO as the sum of TPR and FPR over $y \in {0, 1}$. Presenting TPR and FPR separately could provide a more detailed analysis of the fairness evaluation.
- Baseline methods
   - The proposed method is primarily compared against three baselines. Evaluating it against additional baselines would further strengthen the validation of its effectiveness.
   - Examples:
      - (1) For continuous sensitive attributes: Generalized Demographic Parity, https://openreview.net/pdf/3f9ffe7eafbd44f0205f3629edbcfb60ec738e7c.pdf. Although this work focuses on demographic parity, its approach could be extended to equalized odds.
      - (2) For mixed-type sensitive attributes: Subgroup fairness could also be applicable in this setting, as continuous sensitive attributes can be categorized via binning. https://proceedings.mlr.press/v80/kearns18a.html

**Other Comments Or Suggestions:**

- (Minor typo) Algorithm 2: In the “Input” line, should ($X^{te}, A^{te}, Y^{te}$) be replaced by ($\hat{Y}^{te}, A^{te}, Y^{te}$)?

**Other Strengths And Weaknesses:**

- While the direct generation of pseudo-sensitive attributes through adversarial learning appears to be novel (to the best of my knowledge), adversarial learning-based approaches for developing fair prediction models have been extensively explored in the literature. It would be helpful if the authors could clarify how their proposed method differs from existing adversarial learning-based approaches in terms of intuition, motivation, technical aspects, and experimental results.
- More specifically, how does the generated $\tilde{A}$ differ from the fair representation learned through adversarial learning? It would be helpful if the authors distinguish these two approaches.
   - Several examples are:
      - https://dl.acm.org/doi/pdf/10.1145/3278721.3278779
      - https://arxiv.org/abs/1511.05897
      - https://www.cs.toronto.edu/~toni/Papers/laftr.pdf
      - https://proceedings.mlr.press/v162/kim22b/kim22b.pdf

**Questions For Authors:**

- N/A

**Relation To Broader Scientific Literature:**

- This paper suggests that generating pseudo sensitive attributes can contribute to learning fair prediction models.

**Theoretical Claims:**

- Question: In the proof of Task 2 of Theorem 2.1, the first equation holds when $P(A \le t | Y, S(A) = S) = P (\tilde{A} \le t | Y, S(A) = S),$ while the given result from Task 1 states that $\tilde{A} | Y \overset{d}{=} A | Y$. Could the authors can further explain the math behind how the result of Task 1 directly leads to this equation?

---

> ### Author Rebuttal · Authors · 2025-04-01
>
> We appreciate the feedback by the reviewer. Here are our responses:
>
> 1. Methods: more baselines.
>
>     **Reply.** We thank the reviewer for this valuable suggestion. We add two more baselines in experiments: FDL [4] and Kearns et al. [5].   [Link for experiments added](https://docs.google.com/document/d/1CLwxWwBRsVrYhhIfeC3_IsDBZ0mAoRiZVWWGLv4DDIw/edit?usp=sharing). For ACS income/Adult, we also ran [5] as the reviewer suggests, but we find it failed to converge with limited time given (also evidenced by Figure 1 in its paper). Nonetheless, we agree this could be a good complement and we will update it once it's done.
>
>     We also thank the reviewer for the generalized DP paper ([3]), however, we find that its goal to extend demographic parity is fundamentally different from our goal (equalized odds), and it's also non-trivial to extend it to the best our knowledge. We have included it into our related work section.
>
> 2. Metric:
>
>     (1) Accuracy for classification
>
>     **Reply.** We apologize for the confusion. Our "Loss" metric is misclassification rate for classification task, which is (1 - accuracy) . We will make it clearer in our revised version.
>
>     (2) Why TPR, FPR, or DEO were not mainly used?
>
>     **Reply.** Thank you for this question. The reason why we introduce KPC is that it's a flexible metric for conditional independence without constraints for the shape of input, which well aligns with our goal to measure equalized odds for complex $A$. While DEO (TPR, FPR) is a standard measure in the previous work, it's only suitable for binary $Y$/$A$, thus can only be used in a part of our experiments (Adult/COMPAS).
>
>     (3) Presenting TPR and FPR.
>
>     **Reply.** Thank you for this suggestion. Yes, DEO is a sum of FPR and TPR accross all binary $A$. However, when there are multiple binary $A$, it could be rather subjective to choose which subgroup's TPR(FPR) to be presented, thus making DEO a standard measure ([1, 2]).
>
> 3. Math behind line 583
>
>     **Reply.** We apologize for the confusion. In Task 2, we first prove $\tilde{A} | Y \overset{d}{=} A | Y$, which is done by taking expectation of results from Task 1 ($P(A \leq t \mid Y, S(A)=S)=P(\tilde{A} \leq t \mid Y, S(A)=S)$). We have now explained this in the proof for clarification.
>
> 4. How stable is the adversarial learning process?
>
>     **Reply.** Thank you for this question. We agree that stability could be a potential pitfall for all the adversarial learning approaches, as pointed out in line 432-439. We adopted a vanilla GAN instead of more exquisite ones to provide a fair comparison with FDL [4], but our ICP can integrate with the more efficient methods in the area of adversarial learning. We have now clarified it further in the discussion section.
>
> 5. How our method differs from existing adversarial learning-based approaches.
>
>     **Reply.** Thanks for raising this point.
>
>     While we adopted adversarial learning (Algorithm 1) as in FDL [4], the main contribution of our work is a better way of generating synthetic $\tilde A$ for equalized odds with multi-dimensional (mixed) $A$ (line 090-096).  FDL does not discuss how to estimate $q(A \mid Y)$ when $A$ is complex, which is important for fair ML. Also, the nature of conditioning for equalized odds makes it more difficult than the demographic parity setting (e.g., prior work using VAE [6]). As far as we are aware, no other proposal has been designed for our setting.
>
>     The ICP sampling scheme guides both training and evaluating equalized odds (Algorithm 2). The reported KPCs reflect the same trend of the hypothesis testing (Algorithm 2) when the ground truth is known (Figure 3). This result is useful as ground truth is unknown in practice, which makes results from Algorithm 2 subject to density estimation while KPC itself is not (line 354-360).
>
> 6. How does the generated $\tilde A$ differ from the fair representation learned through adversarial learning?
>
>     **Reply.** Thank you for this question and providing the literature. Learning fair representation is another line of work based on adversarial learning, which generally requires model to predict $A$ from the representation $Z$ (and possibly $Y$) (e.g., the work reviewer mention). When facing complex $A$, this direct way of modelling $A$ could run into the same challenge as FDL [4] does, while our FairICP avoids it as discussed. We thank you again for this suggestion and we have included this discussion into our related work part.
>
>
> ## Reference
>
> [1] Cho et al. A fair classifier using kernel density estimation.
>
> [2] Agarwal et al. A reductions approach to fair classification.
>
> [3] Jiang et al. "Generalized demographic parity for group fairness."
>
> [4] Romano et al. Achieving equalized odds by resampling sensitive attributes.
>
> [5] Kearns et al. Preventing Fairness Gerrymandering: Auditing and Learning for Subgroup Fairness.
>
> [6] Creager et al. "Flexibly fair representation learning by disentanglement."

---

> > ### Comment · Reviewer_NoT1 · 2025-04-02
> >
> > Thank you for the response.
> > I have checked the additional experimental results.
> > However there are a few remaining concerns:
> >
> > -  About points 4-6:
> > I think that the authors should point out a concrete advantage of FairICP compared to other adversarial learning-based methods (that are not designed with sampling pseudo samples).
> > For example, the first reference (https://dl.acm.org/doi/pdf/10.1145/3278721.3278779) is considered as a baseline for FDL, as shown in the FDL paper
> > (Note that I mistakenly referred to this work as a fair representation learning method, but it is actually a direct debiasing method).
> > Further, what are examples of the *"same challenge as FDL [4]"* that existing adversarial learning-based learning would face?
> >
> > - A concern in "Claims And Evidence" seems not to have been discussed in the rebuttal.
> > *However, while the authors claim that the proposed method offers a superior fairness-accuracy trade-off, it seems not consistently outperforms the baselines when the prediction model is linear.*
> >
> > - (Minor) I agree that DEO is a more standard measure, however, it would be better to report the TPR and FPR at least for Adult and COMPAS datasets.

---

> > > ### Author Response · Authors · 2025-04-05
> > >
> > > Thank you for your response! We are happy to address your remaining concerns:
> > >
> > > 1. Compare with other adversarial learning-based methods
> > >
> > >     **Reply.** Thank you for this follow-up question. It's true that Zhang et al. [1] is a direct debiasing method using adversarial learning (similar work [2]), however, they use adversarial network to directly predict $A$ from $\hat Y$ and $Y$ (while in [2] they train a parametric model of $A | \hat Y$ with $\hat Y$ as inputs), which is fundamentally different from our ICP scheme that does not rely on modelling $A | (Y, \hat Y)$. This way of modelling $A | (Y, \hat Y)$ (or similarly modelling $A | Z$ where $Z$ is a representation [3, 4, 5] as the reviewer mentioned), will share a similar challenge as estimating the density of $A|Y$ in FDL when $A$ becomes complex as we discussed as our motivation (line 110-125), e.g., for Zhang et.al [1], how should one balance the prediction loss for $A$ when $A$ is mixed (categorical and continuous)? On the other hand, our FairICP scheme avoid such challenges since we can leverage all the well-studied methods to model $Y | A$ instead, where $Y$ is usually one-dimension (in Figure 2). To empirically support our claims, we also add Zhang et al. [1] as a baseline for the COMPAS dataset ([link](https://docs.google.com/document/d/1iCAwVjXYaUOwlIRMyX_uUXdz5AmBgyEWtuQdUmvs_LM/edit?usp=sharing)).
> > >
> > > 2. "Not outperforming baselines in linear model"
> > >
> > >     **Reply.** We apologize for accidentally dropping our response to this question. As we were plotting loss against KPC/power, we still observed that our model outperforms others in most datasets for linear models (Figure 9, 10). Only in the COMPAS data, which considers binary $A$ and response $Y$, FairICP did not outperform the baselines. However, COMPAS represents a more classical setting with both binary sensitive attributes and response, where Reduction [6] is specifically designed for this setting with an almost oracle calculation of HGR obtained easily [7]. The COMPAS results indicate that FairICP is comparable to Reduction/HGR in cases where their high-quality calculations are feasible. Indeed, even in Figure 4, which uses a non-linear NN structure, FairICP is only comparable to Reduction and slightly better than HGR.  In summary, although COMPAS favors the baseline models by design, we feel it is important to also include it to demonstrate the general applicability of FairICP which is not much worse even if it uses a general framework not specific to COMPAS.
> > >
> > >
> > >
> > > 3. TPR and FPR
> > >
> > >     **Reply.** We agree with the reviewer that reporting TPR and FPR will be a good complement, thus we also report the TPR/FPR for Adult/COMPAS dataset ([link](https://docs.google.com/document/d/1iCAwVjXYaUOwlIRMyX_uUXdz5AmBgyEWtuQdUmvs_LM/edit?usp=sharing)).
> > >
> > > [1] Zhang et al. "Mitigating unwanted biases with adversarial learning." Proceedings of the 2018 AAAI/ACM Conference on AI, Ethics, and Society. 2018.
> > >
> > > [2] Louppe et al. "Learning to pivot with adversarial networks."
> > >
> > > [3] Edwards et al. "Censoring representations with an adversary."
> > >
> > > [4] Madras et al. Learning Adversarially Fair and Transferable Representations.
> > >
> > > [5] Kim et al. "Learning fair representation with a parametric integral probability metric."
> > >
> > > [6] Agarwal et al. A reductions approach to fair classification.
> > >
> > > [7] Mary et al. "Fairness-aware learning for continuous attributes and treatments." International conference on machine learning.

---

### Official Review · Reviewer_pxLZ · 2025-03-14

**Overall Recommendation:** 3

**Summary:**

The paper introduces FairICP, a method for enforcing equalized odds fairness in machine learning models that handle multiple sensitive attributes. The key idea is to improve how synthetic versions of sensitive attributes are generated in fairness-aware learning. Instead of relying on traditional resampling or conditional permutations, FairICP estimates the relationship between the outcome and the sensitive attribute to create better-conditioned synthetic attributes.

The paper compares FairICP to existing methods such as Fair Dummies Learning (FDL) and Conditional Permutation (CP), using Kernel Partial Correlation as a fairness metric. Using a number of experiments, relying on both synthetic data and widely used datasets, the paper argues that FairICP is more effective at reducing fairness violations while maintaining predictive accuracy.

**Claims And Evidence:**

The core claims of the paper are that ICP is a superior method for approximating the true conditional distribution and that this better sampling method leads to a better fairness-accuracy tradeoff (which is the ultimate goal).

Not being a theorist it is hard for me to fully evaluate the theoretical merits of the analysis but I do think that the authors could have put more effort into intuition building. For example, a key element of the analysis is the metric for measuring fairness- kernel partial correlation (KPC). While the authors define the metric mathematically, they do not explain why KPC is an appropriate metric for fairness in this context. Even a simple graphical intuition would have been helpful in understanding what this metric is capturing. It would have been helpful to have a toy example that could concretely demonstrate why CP struggles in a way that ICP does not, or why FDL might introduce bias.

**Essential References Not Discussed:**

I do not have a specific reference that was not included

**Experimental Designs Or Analyses:**

See above for my discussion of experiments.

**Methods And Evaluation Criteria:**

The paper evaluates the method on widely used datasets like COMPAS and ACSIncome, which facilitates comparison with previous work. However, the various choices over which approaches to compare are somewhat opaque. For example, why do the empirical examples not compare ICP to CP? Why is FDL not considered consistently? Why is there a comparison to HGR, which is only very briefly mentioned in the theoretical discussion? The reduction method is not discussed before being used in the example.

Related to the point above, what are the alternatives to the KPC fairness metric? How sensitive are the results of the use of KPC?

**Other Comments Or Suggestions:**

To reiterate the point above, I think it would really improve the paper if the authors spent more time building up intuition for the theoretical claims, the superiority of their claims, and their choice of evaluation metrics.

In analyzing the evaluation of their method, the authors might also consider other methods for equalizing odds, like post-processing, and how they compare with respect to the fairness-accuracy tradeoff. While their choice to focus narrowly on equalizing odds rather than other fairness metrics, it would be beneficial to compare other approaches to equalizing odds.

I also suggest clarifying the various choices made with respect to the real-world experiments, particularly because the comparison to other methods does not fully align with the methods discussed in the previous theoretical section.

**Other Strengths And Weaknesses:**

The biggest strength of the paper is the importance of the issue it is addressing. It has always been a fundamental weakness of the algorithmic fairness literature that it focuses on fairness within one dimension of a protected characteristic when in many cases there are multiple such characteristics that often intersect. Similarly, the choice of the article to focus on equalized odds, although not explicitly discussed, is reasonable given the importance of considering how predictions vary by group, conditional on real outcomes.

Given the potential policy significance of creating an easily implementable method, it is unfortunate the paper is not written in a more accessible way and that it spends little time explaining the intuition for its choices of metrics, superiority of methods and shortcomings of other methods.

**Questions For Authors:**

To summarize the points above, my main questions/issues are:
1. Why do the real world experiments compare ICP to some methods and not others?
2. How does ICP compare to post-processing methods?
3. Is there a way to make the paper more accessible by building up intuition and explaining evaluation choices?

**Relation To Broader Scientific Literature:**

The paper relates to a core challenge within the algorithmic fairness literature of how to treat fairness when sensitive attributes are multidimensional.

**Theoretical Claims:**

I did not check for the correctness of appendix proofs.

---

> ### Author Rebuttal · Authors · 2025-04-01
>
> We appreciate your thorough review and constructive feedback on our paper, here are our responses:
>
> 1. Explanation on KPC.
>
>     **Reply.** Thank you for this constructive feedback. KPC [1] is a recently proposed non-parametric measure for conditional independence without constraints on the shape of inputs. The reason why we choose $KPC(U,V|W)$ aligns with our main goal in this paper: enforcing equalized odds (a conditional independence notion) for multiple $A$ (i.e., making $KPC(\hat{Y}, A|Y)$ small). Additionally, $KPC(U, V|W)$ has been well normalized for direct comparison across different models (supported by our experiments (line 354-360)), and it can be viewed as a generalization of Partial Correlation between $U|W$ and $U|W,V$.
>
>     To see this, we first recall that MMD distance $MMD(U, V)=||\mu_{U}-\mu_{V}||_{\mathcal{H}}^2$ where $\mathcal{H}$ is RKHS and $\mu_U = \int k(,;u)d P_U(u)$ is the kernel mean embedding of a distribution $P\_U(.)$ [9] and consider the simple case where the kernel is linear and $U = \alpha W + \beta V+\varepsilon$ with $W, V, \varepsilon$ being i.i.d $\mathcal{N}(0, 1)$. We then have $P\_{U|W,V}=\mathcal{N}(\alpha W+\beta V, 1)$,  $P\_{U|W}=\mathcal{N}(\alpha W, 1+\beta^2)$ and $P\_{U}=\mathcal{N}(0, 1+\beta^2)$. Then, the numerator in $KPC(U,V|W)$ becomes: $E[MMD^2(P\_{U|W,V}, P\_{U|W})]=E[(\alpha W+\beta V-\alpha W)^2]=\beta^2$. The denominator becomes: $E[MMD^2(\delta\_{U}, P\_{U|W})]=E[(U-\alpha W)^2]=\beta^2+1$.
>
>     Hence, we can see that $KPC(U, W, V)$ reduces to the squared correlation between $U|V$ and $U|W,V$: $KPC(U, W, V)=\rho^2_{U,W|V}$. In this simple case, $U$ is conditionally independent of $V$ given $W$ iff the partial correlation/KPC is 0 ($\beta$ = 0).
>
>     KPC is not the only choice in our setting, and one can choose other conditional dependence measures that allow multi-dimensional inputs (e.g., CODEC [2], though it can be viewed as a special case of KPC). In our paper, the robustness of KPC is evidenced by comparing the similar curve given by other metrics (power of hypothesis testing and DEO, figure 4/7/8 and table 1).
>
>     We thank the reviewer again for this valuable question and we have included some of the intuition in our revised paper.
>
> 2. "Why the experiments not compare ICP to some methods?"
>
>     **Reply.** We apologize for this confusion. This is due to the differences in their applicability: HGR [3] is designed for univariate continuous $A$ and Reduction [4] is only for binary classification and binary attributes. We introduce HGR [3] (in line 105-109) and detail how we generalize it to multiple attributes (line 436-439). We cite Reduction [4] in line 083 and detail it in line 422-425. We now have made it clearer in our revised paper.
>
>     We have also added two more baselines in experiments: FDL [7] and Kearns et al. [10] (introduced in line 096).   [Link for experiments added](https://docs.google.com/document/d/1CLwxWwBRsVrYhhIfeC3_IsDBZ0mAoRiZVWWGLv4DDIw/edit?usp=sharing). We also ran [10] for Adult but find it failed to converge with limited time given (also evidenced by Figure 1 in its paper). Nonetheless, we agree this could be a good complement and we will update it once it's done.
>
>
> 3. "Consider post-processing"
>
>     **Reply.** Thank you for raising this interesting direction. As post-processing and in-processing conventionally represents two different paradigms: the former focuses on training $f(X)$, whereas the latter focuses on recalibrating the probability cutoffs across binary $A$ for binary response (e.g., [6]) and requires test-time access to $A$. Thus, we view it as complementary rather than competing with each other. In fact, one recent work by Tifrea et al. [8], as the rare post-processing work that can work with both a categorical/continuous $A$, proposed to post-process any baseline model, either ERM or in-processing ones.
>
>     Like many in-processing methods, existing post-processing ones are not designed for complex $A$, especially under the equalized. This also raises the question if the idea of ICP can be also utilized in post-processing. We now have included these discussions as future directions in our discussion section.
>
> ## Reference
>
> [1] Huang et al. "Kernel Partial Correlation Coefficient---a Measure of Conditional Dependence."
>
> [2] Azadkia et al. A simple measure of conditional dependence. The Annals of Statistics
>
> [3] Mary et al.. Fairness-aware learning for continuous attributes and treatments.
>
> [4] Agarwal et al. A reductions approach to fair classification.
>
> [6] Kim et al.: Black-Box Post-Processing for Fairness in Classification.
>
> [7] Romano et al. Achieving equalized odds by resampling sensitive attributes.
>
> [8] Tifrea et al. "Frappé: A group fairness framework for post-processing everything."
>
> [9] Tolstikhin et al. "Minimax estimation of maximum mean discrepancy with radial kernels."
>
> [10] Kearns et al. Preventing Fairness Gerrymandering: Auditing and Learning for Subgroup Fairness.

---

> > ### Comment · Reviewer_pxLZ · 2025-04-05
> >
> > I appreciate the clarifications regarding the experiment.

---

> > > ### Author Response · Authors · 2025-04-08
> > >
> > > Thank you for carefully reading our paper and sharing your questions and comments. We hope we have addressed your concerns, and we would be happy to discuss further if any questions remain.

---

### Official Review · Reviewer_Vhcm · 2025-03-18

**Overall Recommendation:** 2

**Summary:**

The authors tackle the problem of enforcing equal odds during training, where the protected attribute is multi-dimensional. They propose FairICP, which constructs a synthetic sensitive attribute via a procedure which involves sampling from a learned $q(Y \mid A)$. Then, a discriminator network is added to distinguish between the real and fake $(Y, A, \hat{Y})$, and the model is penalized if these are distinguishable. The authors evaluate their method against the baselines and find that it achieves a better Pareto front.

## update after rebuttal

I have gone through the rebuttal and the comments from other reviewers. I thank the authors for their additional experiments. However, I believe the lack of novelty (W1) is still a significant weakness that I am not satisfied with. Further, I still believe that the potential application of the method is limited (W2), as it is only useful in cases where the A is high dimensional enough that there is a significant difference in estimation of Y|A vs A|Y, but not high dimensional enough that both fail (and where Kearns et al., (2018) would presumably be better). Finally, as also raised by other reviewers, the performance/fairness improvements are not consistent over the baselines, especially in the rebuttal results. For these reasons, I will keep my score.

**Claims And Evidence:**

1. The novelty of the paper over Romano et al. (2020) is rather limited. Algorithm 2 is identical to this prior work, and the overall architecture (and Algorithm 1) are essentially the same as this prior work with a small tweak to the sampling which is a simple application of Bayes rule. As such, I do not believe it presents a sufficiently novel algorithmic or theoretical contribution.

2. The main motivation of this work is to allow for equal odds when the attribute is multi-dimensional. However, as the dimension of A grows, the estimation of $Y \mid A$ will become worse due to smaller per-group sample size. This seems to limit the utility of the method, and the authors have not characterized this theoretically.

3. The authors state (L94) that "While several studies have successfully addressed demographic parity under multiple sensitive attributes via permutations(Kearns et al., 2018; Creager et al., 2019), these ideas can not be trivially generalized to equalized odds since permutation of A conditional Y is still difficult and does not circumvent the challenge in distribution estimation of multidimensional complex attributes". I do not believe this is true for Kearns et al., (2018), as they specifically examine equalizing FPR in their paper (and equalizing FNR is also possible by symmetry); and do not require estimating the conditional density. The authors should add this method as a baseline.

**Essential References Not Discussed:**

Please see above.

**Experimental Designs Or Analyses:**

Please see "Methods And Evaluation Criteria".

**Methods And Evaluation Criteria:**

1. The Adult and COMPAS datasets are not great testbeds for the method as the number of groups (intersections of sex and race) is only 4. Thus, given that these are also binary classification problems, it is possible to estimate the conditional probability distribution by data analyses (without training any models), and any in-processing equal odds method would work here. For these datasets, the authors should at least show FDL as a baseline. They should also compare against the line of work on fairness via adversaries (e.g. [1]), which has a similar architecture but does not require conditional randomization.

2. The author should show ERM as a single point on all Pareto plots.


[1] Learning Adversarially Fair and Transferable Representations. ICML 2018.

**Other Comments Or Suggestions:**

N/A

**Other Strengths And Weaknesses:**

N/A

**Questions For Authors:**

1. Empirically, the DEO is still quite large (Figure 8 and Table 1). Is this because DEO is measuring fairness for binary $\hat{Y}$ whereas the method enforces equal odds wrt the continuous score?

**Relation To Broader Scientific Literature:**

The work is one of many that enforces equal odds during supervised learning. However, the paper distinguishes itself by examining the case of multidimensional attributes, where existing methods would not work or are less effective. The proposed method relies heavily on the idea of conditional randomization (Candes et al., 2018), which has been applied to enforce equal odds for univariate attributes in Romano et al. (2020). The main contribution of the paper is an alternative sampling procedure which adapts Romano et al. (2020) to the multi-attribute setting.

**Theoretical Claims:**

I did not check any proofs in detail.

---

> ### Author Rebuttal · Authors · 2025-04-01
>
> We appreciate the feedback from the reviewer. Here are our responses:
>
> 1. Novelty of the paper over FDL.
>
>     **Reply.** Thank you for sharing your concern. We agree that our core insight is not complicated, yet it has not been previously explored and is effective.
>
>     1) While we adopted adversarial learning (Algorithm 1) as in FDL [1], the main contribution of our work is a better way of generating synthetic $\tilde A$ for equalized odds with multi-dimensional (mixed) $A$ (line 090-096).  FDL does not discuss how to estimate $q(A \mid Y)$ when $A$ is complex, which is important for fair ML. Also, the nature of conditioning for equalized odds makes it more difficult than the demographic parity setting (e.g., prior work using VAE [4]). As far as we are aware, no other proposal has been designed for our setting.
>
>         The ICP sampling scheme guides training and can measure equalized odds (Algorithm 2). The reported KPCs reflect the same trend of the hypothesis testing (Algorithm 2) when the ground truth is known (Figure 3). This result is useful as ground truth is unknown in practice, which makes results from Algorithm 2 subject to density estimation, but the KPC itself is not (line 354-360).
>
>     2)  While our proof uses Bayes rule, the ICP procedure has not been proposed before. The simplicity in the proof does not contradict its usefulness. For example, one influential result in the classical density ratio estimation is framed as a classification problem [7]. The validity of this transformation is also just Bayes rule, but doesn’t reduce its novelty or utility.
>
> 2. "The estimation of $Y | A$ gets worse with high-dim A."
>
>     **Reply.** We apologize for the confusion. If the problem is intrinsically hard, e.g., there are $n$ samples with each $(Y_i, A_i)$ being drastically different, even the distribution of $Y|A$ will be poorly estimated. However, estimation of the $(A|Y)$ is not easier in this setting.
>
>     The gain of using $Y|A$ are twofold (line 110-124):
>
>     1) Machine learning approaches can more effectively leverage the structure of $Y | A$ than the direct estimation of $A|Y$. E.g., when $(Y, A)$ are jointly Gaussian with sparse connections, the former can provide better quality from using lasso (MSE bound $\sqrt{\frac{s\log p}{n}}$ under sparsity condition $s$, [e.g.,8]) whereas the latter is prone to more errors even when using graphical lasso (Frobenius bound $~\sqrt{\frac{(s^2+p)\log p}{n}}$, e.g., [6]), which can also be evidenced by our Figures 2. These bounds on density estimation will in turn upper bound the quality of the conditional permutation (Appendix C.1).
>
>     2) Modeling $Y \mid A$ avoids directly estimating complex dependencies in multidimensional $A$ (e.g., continuous and binary variables). Instead, dependence in $A$ is naturally incorporated through $Y \mid A$, simplifying implementation.
>
>     We have now clarified this further in section 2.2.
>
> 3. More baselines
>
>     **Reply.** We thank the reviewer for this valuable suggestion. We add two more baselines in experiments: FDL [1] and Kearns et al. [2].   [Link for experiments added](https://docs.google.com/document/d/1CLwxWwBRsVrYhhIfeC3_IsDBZ0mAoRiZVWWGLv4DDIw/edit?usp=sharing). We also ran [2] for Adult but found it failed to converge with limited time given (also evidenced by Figure 1 in its paper). Nonetheless, we agree this could be a good complement and we will update it once it's done.
>
>     For LAFTR, we find it can only be applied to a single binary $A$, which differs from our main target in this paper.
>
> 5. "The DEO is still quite large."
>
>     **Reply.** We apologize for the confusion. We calculated DEO (line 823) as the sum of TPR and FPR differences as in [9], while some work [10] defines it as the max of differences, which explains why our results appear to be large. Our method still provides generally smaller DEOs, which indicates the consistency of KPC and DEO (Figure 8 and Table 1).
>
> ## References
>
> [1] Romano et al. Achieving equalized odds by resampling sensitive attributes.
>
> [2] Kearns et al. Preventing Fairness Gerrymandering: Auditing and Learning for Subgroup Fairness.
>
> [3] Madras et al. Learning Adversarially Fair and Transferable Representations.
>
> [4] Creager et al. "Flexibly fair representation learning by disentanglement."
>
> [5] Tansey et al. "The holdout randomization test for feature selection in black box models."
>
> [6] Wang et al. "Precision matrix estimation in high dimensional gaussian graphical models with faster rates."
>
> [7] Menon et al. Linking losses for density ratio and class-probability estimation.
>
> [8] Wainwright et al. High-dimensional statistics: A non-asymptotic viewpoint.
>
> [9] Cho et al. A fair classifier using kernel density estimation.
>
> [10] Agarwal et al. A reductions approach to fair classification.

---

### Decision · Program_Chairs · 2025-05-01

**Decision:**

Accept (poster)

**Comment:**

This paper proposes FairICP, an in-processing method for enforcing equalized odds as a fairness notion in supervised learning settings with multi-dimensional and mixed-type sensitive attributes. The central technical contribution is an Inverse Conditional Permutation (ICP) procedure, which uses Bayes rule to generate pseudo-sensitive attributes without directly estimating high-dimensional conditional distributions. The ICP mechanism is coupled with adversarial training to enforce equalized odds. FairICP is evaluated across synthetic and real-world datasets and compared with several recent fairness-aware learning methods.

**Strengths**:
* The paper addresses a non-trivial gap in the fairness literature: enforcing equalized odds when sensitive attributes are multi-dimensional, mixed-type, or not easily binarized.
* The ICP-based sampling approach is a natural yet underexplored extension of prior permutation-based techniques and is simple and practical.
* The authors offer theoretical justification for the advantages of ICP over classical conditional permutation, particularly when density estimation is inaccurate (which can be a realistic scenario).
* The empirical evaluation is broad: synthetic setups and several widely used datasets (Adult, COMPAS, ACSIncome), and includes comparisons with relevant baselines like FDL, Kearns et al., and various adversarial approaches.
* The authors were responsive during the discussion phase and added additional baselines, ablations, and clarity to address reviewer concerns.

**Reviewer discussions**:
* Reviewer pxLZ (Weak Accept) appreciates the importance of the problem and finds the method promising but criticizes the writing and lack of intuition, especially around metrics like KPC. Their review is ultimately cautious but supportive.
* Reviewer NoT1 (Weak Accept) notes that while FairICP is flexible and shows good performance, it does not consistently outperform baselines under linear models. They also request more clarity in comparisons with adversarial learning-based methods. These concerns were addressed in the rebuttal.
* Reviewer Vhcm (Weak Reject) argues that the method is an incremental extension of prior work (notably Romano et al.), and point that the method’s utility is limited to a narrow set of cases and raise concerns about inconsistent performance. These concerns were addressed by the authors (and the reviewer did not engage further).
* Reviewer pguj (Weak Accept) praises the ICP idea and the ability to handle complex sensitive attributes but raises concerns about reliance on density estimation, limited datasets, and whether FairICP can generalize to other fairness criteria. These points were acknowledged and partially addressed.

**Conclusion**:
This is a borderline paper with a solid technical contribution and an important practical motivation. The proposed method is not radically novel in its architecture (building on adversarial fairness and conditional permutation), but the ICP sampling mechanism for complex sensitive attributes does seem a useful and generalizable idea. The authors provide theoretical support and empirical validation. While presentation could be improved and some reviewers remain skeptical about novelty, I find the core contribution meaningful and the paper sufficiently substantiated.